# Stochastic Control for Fine-tuning Diffusion Models: Optimality, Regularity, and Convergence

Yinbin Han [1]   Meisam Razaviyayn [2]   Renyuan Xu [1]

## Abstract

Diffusion models have emerged as powerful tools for generative modeling, demonstrating exceptional capability in capturing target data distributions from large datasets. However, fine-tuning these massive models for specific downstream tasks, constraints, and human preferences remains a critical challenge. While recent advances have leveraged reinforcement learning algorithms to tackle this problem, much of the progress has been empirical, with limited theoretical understanding. To bridge this gap, we propose a stochastic control framework for fine-tuning diffusion models. Building on denoising diffusion probabilistic models as the pre-trained reference dynamics, our approach integrates linear dynamics control with Kullback–Leibler regularization. We establish the well-posedness and regularity of the stochastic control problem and develop a policy iteration algorithm (PI-FT) for numerical solution. We show that PI-FT achieves global convergence at a linear rate. Unlike existing work that assumes regularities throughout training, we prove that the control and value sequences generated by the algorithm preserve the desired regularity. Finally, we extend our framework to parametric settings for efficient implementation and demonstrate the practical effectiveness of the proposed PI-FT algorithm through numerical experiments.

## 1. Introduction

The growing availability of large-scale datasets, coupled with advances in high-performance computing, has fueled the rise of data-driven methods across scientific and engineering fields. Traditional approaches, which fit models to offline or online data, heavily depend on data quality and availability. As a result, data limitations can significantly degrade performance (Doersch, 2016; Durgadevi et al., 2021; Fetaya et al., 2020). For instance, constraints in experimental design often lead to data scarcity, hindering model effectiveness. Moreover, acquiring sufficient data for reliable experimentation is costly and time-consuming, posing scalability challenges. Most critically, data collected under specific conditions may not generalize well to new environments or downstream tasks, limiting the adaptability of learned models (Alzubaidi et al., 2023; Bansal et al., 2022; Gangwal et al., 2024).

Generative modeling offers a flexible solution by creating synthetic data that maintains the properties of collected data while enhancing data diversity. Diffusion models, such as those proposed by Ho et al. (2020); Sohl-Dickstein et al. (2015); Song & Ermon (2019); Song et al. (2021), have emerged as powerful tools in this area, supporting notable advancements such as DALL·E (Betker et al., 2023; Ramesh et al., 2022), Stable Diffusion (Rombach et al., 2022), and Sora (OpenAI, 2024). These models excel by learning the *score function* from potentially high-dimensional and limited data, extracting critical information for the data generation process. To achieve cost-effective, task-specific data generation and multi-modal integration, techniques like alignment (Wallace et al., 2024), including guidance (Dhariwal & Nichol, 2021; Ho & Salimans, 2021) and fine-tuning (Black et al., 2024; Clark et al., 2024; Fan & Lee, 2023; Fan et al., 2024), of pre-trained models are proposed. While diffusion alignment has achieved significant empirical success, the theoretical foundations remain in the early stages—a gap this paper seeks to address.

Aligning a pre-trained diffusion model to a specific task or dataset requires updating its parameters through additional training, typically using a smaller, task-specific dataset (Dai et al., 2023; Podell et al., 2023; Rombach et al., 2022; Wallace et al., 2024). Among the prevalent alignment techniques, fine-tuning and guidance differ in how the additional criteria are handled—either as a soft constraint or a hard constraint. The soft constraint approach is particularly effective for applications that incorporate human preference

*Equal contribution  [1]Department of Finance and Risk Engineering, New York University [2]Daniel J. Epstein Department of Industrial and Systems Engineering, University of Southern California. Correspondence to: Renyuan Xu <rx2364@nyu.edu>.

*Proceedings of the 42$^{nd}$ International Conference on Machine Learning*, Vancouver, Canada. PMLR 267, 2025. Copyright 2025 by the author(s).

or embed human values as a reward signal evaluated on the task-specific dataset (Black et al., 2024; Clark et al., 2024; Fan et al., 2024; Uehara et al., 2024c; Zhao et al., 2024b), which is the central focus of our paper.

**Our work and contributions.**

We introduce a discrete-time stochastic control formulation with linear dynamics and Kullback–Leibler (KL) regularization for fine-tuning diffusion models. Specifically, we establish a novel connection to denoising diffusion probabilistic models (DDPMs) (Ho et al., 2020) by treating the fine-tuned score as a *control to be learned*. In particular, the soft constraint or human preference is modeled by a reward signal evaluated on the generated data output and the KL regularization term penalizes the deviation of the control from the pre-trained score function. We utilize the discrete-time formulation since fine-tuning is *inherently* a *discrete-time* problem, as it relies on a pre-trained model which is typically implemented in discrete time, with DDPM being the most commonly-used example. By appropriately selecting the regularization parameter, we demonstrate the well-posedness of the control problem and analyze the regularity of the optimal value function. A key insight arises from the concavity of the nested one-step optimization problem in the Bellman equation, a direct result of KL regularization under a properly chosen regularization coefficient. Leveraging this property, we develop a policy iteration algorithm (PI-FT) with guaranteed convergence to the globally optimal solution at a linear rate. A central challenge involves preserving the regularity of the sequence of value functions and output controls generated during algorithm iterations. This is achieved through a novel coupled induction argument and precise estimations of the regularization parameters. Finally, we discuss the algorithm design in parametric settings and conduct thorough experiments to demonstrate the practical efficiency and effectiveness of the proposed PI-FT method.

While existing methods such as DPOK (Fan & Lee, 2023) and DDPO (Black et al., 2024) rely on generic RL developments such as PPO or REINFORCE, they do not fully leverage the fine-tuning structure. On the contrary, the specific setting considered in this work, with linear dynamics and entropy-regularization, is tailored towards the development of efficient fine-tuning diffusion models. For this reason, the PI-FT algorithm and its parametric extension directly computes the policy gradient of a KL-regularized control objective. This principled design leads to a more efficient implementation in practice compared to prior works, while also offering theoretical guarantees.

Despite promising empirical evidence in the literature, analyzing algorithm convergence remains challenging due to the continuous nature of the state-action space. To the best of our knowledge, this work is the first on fine-tuning diffusion models that presents a *convergence guarantee*. We address this challenge by leveraging linear dynamics and the (one-step) concavity introduced by the KL regularization term, which inherently captures the essence of fine-tuning problems. Specifically, we establish the universal regularity of both the optimal value function and the sequence of value functions generated *during* iterative updates. Moreoverover, the foundational techniques underlying our results extend naturally to parametric algorithm design, offering insights beyond the specific context of diffusion model fine-tuning.

**Related literature.**

Our work is related to several emerging research directions.

**Fine-tuning of diffusion models.** Our framework is closely related to the recent studies on fine-tuning diffusion models. Motivated by advancements in the (discrete-time) RL literature, Fan & Lee (2023) was the first to introduce a reward-based approach for improving pre-trained diffusion models. Building on this idea, two concurrent works (Black et al., 2024; Fan et al., 2024) proposed a Markov decision process (MDP) formulation for denoising diffusion processes. To mitigate reward over-optimization (Gao et al., 2023), Fan et al. (2024) examined the impact of incorporating KL regularization as an implicit reward signal. Following this idea, Uehara et al. (2024b) introduced an online fine-tuning framework where the regret is upper-bounded by the accuracy of a statistical error oracle for reward estimation. Just recently, Yoon et al. (2024) formulated diffusion training as an inverse RL problem and proposed a value-based algorithm. For a comprehensive review of this topic, we direct interested readers to Uehara et al. (2024a). Beyond RL-based fine-tuning methods, alternative approaches include classifier guidance (Dhariwal & Nichol, 2021) and classifier-free guidance (Ho & Salimans, 2021), supervised fine-tuning (Lee et al., 2023), LoRA and its variants (Hu et al., 2022; Ryu, 2023), DreamBooth (Ruiz et al., 2023), and DRaFT (Clark et al., 2024) among others.

All the aforementioned references focused solely on empirical investigations, except for Uehara et al. (2024b), which leaves theoretical foundations largely unexplored. While the MDP formulation discussed above aligns well with the RL literature, it overlooks the structural properties inherent to diffusion models. In contrast, our approach specifically leverages DDPM as the pre-trained model, thereby fully utilizing the underlying structure of diffusion models. Moreover, the stochastic control formulation is more appropriate than RL as the dynamics are known in diffusion models and exploration is not necessary. Consequently, the efficiency of our algorithm stems from taking control-based formulation.

**Continuous-time control/RL for fine-tuning.** Inspired by the continuous-time nature of diffusion processes, recent

work has explored the application of continuous-time control/RL for fine-tuning. For instance, Berner et al. (2024) established a connection between fine-tuning and stochastic control by analyzing the Hamilton-Jacobi-Bellman (HJB) equation for the log density. Domingo-Enrich et al. (2025) introduced an adjoint matching method for the fine-tuning of rectified flow, drawing inspiration from Pontryagin's maximum principle. A more rigorous treatment of fine-tuning within the framework of entropy-regularized control problems is developed in Tang (2024), addressing the well-posedness and regularity of the corresponding HJB equation and extending the analysis to general $f$-divergences, though no specific algorithm was proposed. Following this entropy-regularization perspective, Zhao et al. (2024a) derived the formulas for continuous-time policy gradient method and continuous-time version of proximal policy optimization (PPO). Furthermore, Gao et al. (2024) applied a continuous-time q-learning algorithm to simulate data that reflects human preferences directly, without relying on pre-trained models.

Compared with the literature that technically remain at the level of exploring HJB equations, we establish convergence rate for the proposed algorithm in discrete time. Despite the continuous-time nature of diffusion processes, the discrete-time setting is more suitable because fine-tuning builds upon a pre-trained model, which is naturally implemented and provided in discrete time.

**RL theory.** Our theoretical developments are also closely connected to the literature of RL and policy gradient methods in discrete time. For MDPs with *finite* state and action spaces, recent advancements providing global convergence guarantees for policy gradient methods and their variants can be found in Berner et al. (2024); Bhandari & Russo (2021; 2024); Cen et al. (2021); Ding et al. (2020); Fatkhullin et al. (2023); Fu et al. (2021); Liu et al. (2019; 2020); Mondal & Aggarwal (2024); Wang et al. (2020b); Xiao (2022); Xu et al. (2021); Zhan et al. (2023); Zhang et al. (2020; 2021). Beyond MDPs, policy gradient methods have also been applied to Linear Quadratic Regulators (LQRs), a specific class of control problems characterized by linear dynamics and quadratic cost functions (Bu et al., 2019; Fazel et al., 2018; Guo et al., 2023; Hambly et al., 2021; Han et al., 2023; Malik et al., 2019; Mohammadi et al., 2019; Szpruch et al., 2021; 2024; Wang et al., 2020a; Zhou & Lu, 2023). While the convergence analysis of policy gradient methods is well-established in the above-mentioned settings, the study on control problems with continuous state-action space and general cost functions has been limited. Our work ventures into this broader class of control problems, providing convergence guarantees for policy gradient algorithms under more general settings.

**Notations and organization.**

Take $d \in \mathbb{N}_+$, the set of all positive integers. We denote $(\Omega, \mathcal{F} := \{\mathcal{F}_t\}_{t \geq 0}, \mathbb{P})$ as a usual filtered probability space supporting a random variable $R \in L^1(\mathbb{R}^d, \mathcal{F}_0)$, namely, $R(y) \in \mathcal{F}_0$ has finite mean for all $y \in \mathbb{R}^d$. Denote $\{W_t\}_{t=0}^{T-1}$ a sequence of independent standard $d$-dimensional Gaussian random vectors. We denote by $\mathbb{F}^W$ the natural filtration of $\{W_t\}_{t=0}^{T-1}$. In addition, $\mathcal{N}(\mu, \Sigma)$ denotes the normal distribution with mean $\mu \in \mathbb{R}^d$ and covariance $\Sigma \in \mathbb{R}^{d \times d}$. Let $f(\cdot | \mu, \Sigma)$ denote the probability density of $\mathcal{N}(\mu, \Sigma)$. For two probability distributions $P$ and $Q$ such that $P \ll Q$ with densities $p$ and $q$ supported on $\mathbb{R}^d$, we denote the KL divergence by $\mathrm{KL}(p\|q) := \int_{\mathbb{R}^d} p(y) \log \frac{p(y)}{q(y)} \mathrm{d}y$. Finally, $I_d \in \mathbb{R}^{d \times d}$ denotes the identity matrix and $\|\cdot\|_2$ denotes the Euclidean norm.

The paper is organized as follows. Section 2 introduces the set-up of the stochastic control problem and establishes the well-posedness and regularity. Section 3 proposes a policy iteration algorithm, develops the linear convergence result, and discusses a parametric extension. Finally, numerical experiments are demonstrated in Section 4.

## 2. Problem set-up and regularity results

### 2.1. Problem set-up

To address the challenge of fine-tuning, we first introduce the dynamics of denoising diffusion probabilistic models (DDPMs), a widely adopted pre-trained framework in practice. Our focus is on the discrete-time formulation, which aligns with the standard implementation of diffusion models in practice. In addition, DDPMs serve as a foundational scheme for effective fine-tuning.

**Denoising diffusion probabilistic models (DDPM).** A well-trained DDPM $\{Y_t^{\mathrm{pre}}\}_{t=0}^T$ in discrete time follows the following stochastic dynamics with state $Y_t^{\mathrm{pre}} \in \mathbb{R}^d$:

$$Y_{t+1}^{\mathrm{pre}} = \frac{1}{\sqrt{\alpha_t}}\big(Y_t^{\mathrm{pre}} + (1 - \alpha_t)s_t^{\mathrm{pre}}(Y_t^{\mathrm{pre}})\big) + \sigma_t W_t, \quad (1)$$

with $Y_0^{\mathrm{pre}} \sim \mathcal{N}(0, I_d)$. Here, $\{W_t\}_{t=0}^{T-1}$ are i.i.d. standard Gaussian random vectors such that $W_t \sim \mathcal{N}(0, I_d)$. The hyper-parameters $\{\alpha_t\}_{t=0}^{T-1}$ with $\alpha_t \in (0, 1)$ and $\{\sigma_t\}_{t=0}^{T-1}$ with $\sigma_t > 0$ represent the prescribed denoising rate schedules (Ho et al., 2020; Li et al., 2024). They control the variance of noise in data generation[1]. Here, $s_t^{\mathrm{pre}} : \mathbb{R}^d \to \mathbb{R}^d$ is the score function associated with the pre-trained model. In practice, the score estimator $s_t^{\mathrm{pre}}$ is obtained by training a neural network to minimize the score matching loss (Hyvärinen & Dayan, 2005). With a well-trained pre-trained

---

[1]While DDPM chooses $\sigma_t^2 = 1/\alpha_t - 1$, our analysis in the subsequent sections works for general $\sigma_t$.

model, we expect the distribution of $Y_T^{\mathrm{pre}}$ to be close to the target distribution, from which the pre-trained model has access to samples and seeks to generate additional samples. Given $s_t^{\mathrm{pre}}$, we denote $p_{t+1|t}^{\mathrm{pre}}(\cdot|y_t)$ as the conditional density of $Y_{t+1}^{\mathrm{pre}}$ given $Y_t^{\mathrm{pre}} = y_t$ induced by the dynamics (1), i.e.,

$$p_{t+1|t}^{\mathrm{pre}}(\cdot|y_t) = f\left(\cdot\left|1/\sqrt{\alpha_t}\left(y_t + (1-\alpha_t)s_t^{\mathrm{pre}}(y_t)\right), \sigma_t^2 I_d\right.\right).$$

*Remark* 2.1 (Choice of the pre-trained model.). DDPMs have become the leading choice for pre-trained models across a wide range of applications, serving as a powerful building block for fine-tuning in diverse tasks (Dhariwal & Nichol, 2021; Ho et al., 2020; Li et al., 2024; Watson et al., 2023). By pre-training on large datasets, DDPMs capture complex data distributions, enabling relatively easier fine-tuning for specialized applications such as conditional image synthesis, text-to-image generation, and even time-series forecasting. This versatility has been demonstrated in models such as Stable Diffusion (Rombach et al., 2022) and DALL·E (Ramesh et al., 2022), where fine-tuning on task-specific data enhances performance and customization (Fan et al., 2024).

**Stochastic control formulation.** When human preferences or soft constraints are modeled through a stochastic reward function $R(\cdot)$, our objective is to maximize the following stochastic control formulation, which models fine-tuning tasks in diffusion models:

$$\mathbb{E}\left[R(Y_T) - \sum_{t=0}^{T-1} \beta_t \mathrm{KL}\left(p_{t+1|t}(\cdot|Y_t) \,\middle\|\, p_{t+1|t}^{\mathrm{pre}}(\cdot|Y_t)\right)\right], \quad (2)$$

with state $Y_t \in \mathbb{R}^d$ and control $U_t \in \mathbb{R}^d$ following the dynamics

$$Y_{t+1} = \frac{1}{\sqrt{\alpha_t}}\left(Y_t + (1-\alpha_t)U_t\right) + \sigma_t W_t, \quad (3)$$

where $Y_0 \sim \mathcal{N}(0, I_d)$. In particular, we work with Markovian policies such that the objective in (2) is well-defined. Mathematically, the control process $\{U_t\}_{t=0}^{T-1}$ is admissible if each $U_t$ can be expressed as $U_t = u_t(Y_t)$, where $u_t : \mathbb{R}^d \to \mathbb{R}^d$ is a measurable function ensuring that the objective in (2) remains finite. In terms of the reward, we assume $R \in L^1(\mathbb{R}^d, \mathcal{F}_0)$, namely, $R(y) \in \mathcal{F}_0$ has finite mean for all $y \in \mathbb{R}^d$. We assume the reward function is *known* in this work. The KL-divergence measures the deviation of the fine-tuned model $p_{t+1|t}$ from the pre-trained model $p_{t+1|t}^{\mathrm{pre}}$, where $p_{t+1|t}$ denotes the conditional distribution of $Y_{t+1}$ given $Y_t$ induced by the control policy $u_t$. The regularization coefficients $\{\beta_t\}_{t=0}^{T-1}$ control the strength of the regularization term.

We have a few remarks in place.

*Remark* 2.2 (Control as score function). Our formulation is rooted in both diffusion models and control theory. Comparing (3) with (1), we replace the pre-trained score $s_t^{\mathrm{pre}}$ by a control policy $u_t$. Consequently, in the context of diffusion models, the learned control sequence $\{u_t\}_{t=0}^{T-1}$ can be viewed as the *new* score function of the fine-tuned model. In other words, solving the control problem (2)–(3) is essentially learning a new score function in response to the reward signal $R(\cdot)$ for fine-tuning. Note that the linear dynamics (3) with Gaussian noise is preferred in stochastic control due to its tractability in analysis. Specifically, our theoretical analysis in the subsequent sections heavily relies on the linearity and the Gaussian smoothing effect.

*Remark* 2.3 (Rationale for the objective function and the known reward assumption). The objective function in the control formulation (2)–(3) consists of two parts: a terminal reward function $R(\cdot)$ at time $T$ and intermediate KL penalties. The reward $R(\cdot)$ captures human preference on the generated samples. For example, in the text-to-image generation, the reward $R(\cdot)$ represents how the generated data $Y_T$ is aligned with the input prompt (Black et al., 2024; Fan et al., 2024). In addition, the penalty term ensures that the fine-tuned model is not too far away from the pre-trained model, which prevents overfitting. Unlike the Shannon entropy of the (randomized) control policy commonly used in the RL literature to encourage exploration, the KL regularization in (2) is applied between two conditional probability densities. From an optimization perspective, the KL divergence introduces concavity in control and consequently leads to a better landscape of the objective. Indeed, we will choose the parameter $\beta_t$ sufficiently large to guarantee the existence and uniqueness of the optimal control and to satisfy certain regularity conditions; see Theorem 2.8.

In practice, fine-tuning is typically performed on a dataset containing reward ratings for each data sample, which is much smaller than the pre-trained dataset. Using this fine-tuning dataset, one can estimate the expected reward function. The assumption of known reward is not overly restrictive, as online learning to acquire new rewards is not generally uncommon.

*Remark* 2.4 (Connection to KL divergence on path-wise measures.). Another natural idea is to impose the KL divergence between the path-wise measures, i.e., $\mathrm{KL}(p_{0:T}\|p_{0:T}^{\mathrm{pre}})$, where $p_{0:T}$ is the joint density of $\{Y_t\}_{t=0}^T$ and $p_{0:T}^{\mathrm{pre}}$ is the joint density of $\{Y_t^{\mathrm{pre}}\}_{t=0}^T$. This choice is relevant to formulation (2)–(3). In particular, the Markov property of the dynamics and the chain rule of the KL divergence imply

$$\mathrm{KL}(p_{0:T}\|p_{0:T}^{\mathrm{pre}}) = \mathbb{E}\left[\sum_{t=0}^{T-1} \mathrm{KL}\left(p_{t+1|t}(\cdot|Y_t)\|p_{t+1|t}^{\mathrm{pre}}(\cdot|Y_t)\right)\right],$$

where the expectation is taken over all random variables

$\{Y_t\}_{t=0}^T$. Here, we define

$$\mathrm{KL}(p_{0:T}\|p_{0:T}^{\mathrm{pre}}) := \int p_{0:T}(\mathbf{y}) \log \frac{p_{0:T}(\mathbf{y})}{p_{0:T}^{\mathrm{pre}}(\mathbf{y})}\,d\mathbf{y}, \quad (4)$$

with $\mathbf{y} = (y_0, \ldots, y_T)$, and for given $y_t \in \mathbb{R}^d$, denote

$$\mathrm{KL}\Big(p_{t+1|t}(\,\cdot\,|y_t)\|p_{t+1|t}^{\mathrm{pre}}(\,\cdot\,|y_t)\Big)$$
$$:= \int p_{t+1|t}(y_{t+1}|y_t) \log \frac{p_{t+1|t}(y_{t+1}|y_t)}{p_{t+1|t}^{\mathrm{pre}}(y_{t+1}|y_t)}\,dy_{t+1}. \quad (5)$$

Thus, when $\beta_t = \beta$ for all $t$, the objective in (2) becomes

$$\mathbb{E}\left[R(Y_T) - \sum_{t=0}^{T-1} \beta_t \mathrm{KL}\Big(p_{t+1|t}(\cdot|Y_t)\|p_{t+1|t}^{\mathrm{pre}}(\cdot|Y_t)\Big)\right]$$
$$= \mathbb{E}\left[R(Y_T)\right] - \beta \mathrm{KL}(p_{0:T}\|p_{0:T}^{\mathrm{pre}}).$$

KL divergence is a common choice of regularization in fine-tuning of diffusion models (Fan et al., 2024; Gao et al., 2024; Uehara et al., 2024b; Zhao et al., 2024a). However, our work is the first to use KL divergence over *transition dynamics (on path space)* to control the deviation of the fine-tuned model from the pre-trained model, whereas formulations in the literature consider KL between the *terminal state distributions*. Our formulation potentially utilizes more information from the rich pre-trained models.

Given two Gaussian densities $p_{t+1|t}$ and $p_{t+1|t}^{\mathrm{pre}}$ in (2), the following lemma simplifies the KL divergence term.

**Lemma 2.5.** *Consider the KL divergence defined as in* (5). *For any $y_t \in \mathbb{R}^d$ and any admissible control policy $u_t$, it holds that*

$$\mathrm{KL}\Big(p_{t+1|t}(\cdot|y_t)\|p_{t+1|t}^{\mathrm{pre}}(\cdot|y_t)\Big)$$
$$= \frac{(1-\alpha_t)^2}{2\alpha_t\sigma_t^2} \|u_t(y_t) - s_t^{\mathrm{pre}}(y_t)\|_2^2. \quad (6)$$

Lemma 2.5 links the KL-divergence between two conditional distributions with the squared loss between the control $u_t$ and the pre-trained score $s_t^{\mathrm{pre}}$. Since the control can be interpreted as the new score of the fine-tuned model, (6) enjoys the spirit of the score matching loss in training diffusion models (Han et al., 2024; Ho et al., 2020; Song et al., 2021). The proof of Lemma 2.5 is based on direct calculations; see Appendix B. With this in hand, we define the optimal value function at time $t$ as

$$V_t^*(y) := \sup \mathbb{E}\left[-\sum_{\ell=t}^{T-1} \beta_\ell \frac{(1-\alpha_\ell)^2}{2\alpha_\ell\sigma_\ell^2} \|u_\ell(Y_\ell) - s_\ell^{\mathrm{pre}}(Y_\ell)\|_2^2 \right.$$
$$\left. + R(Y_T) \,\Big|\, Y_t = y\right], \quad (7)$$

where the supremum is taken over all admissible control policies $\{u_\ell\}_{\ell=t}^{T-1}$. The Dynamic Programming Principle implies that $V_t^*$ satisfies the Bellman equation:

$$V_t^*(y) = \sup_{u_t} \mathbb{E}\left[V_{t+1}^*\Big(\frac{1}{\sqrt{\alpha_t}}\left(y + (1-\alpha_t)u_t(y)\right) + \sigma_t W_t\Big)\right.$$
$$\left. - \beta_t \frac{(1-\alpha_t)^2}{2\alpha_t\sigma_t^2} \|u_t(y) - s_t^{\mathrm{pre}}(y)\|_2^2\right]. \quad (8)$$

In the next subsection, we will discuss the well-posedness of the optimal control problem (2)–(3) and the regularity of the optimal value function $V_t^*$.

## 2.2. Regularity and well-posedness

In this subsection, we establish the regularity of the optimal value function and the optimal control policy. To start, we make the following assumptions on the reward and pre-trained score functions.

**Assumption 2.6** (Smoothness of the reward). Assume $r(y) := \mathbb{E}[R(y)]$ is $L_0^r$-Lipschitz and $L_1^r$-gradient Lipschitz in $y \in \mathbb{R}^d$, i.e., the following holds for any $y_1, y_2 \in \mathbb{R}^d$:

$$|r(y_1) - r(y_2)| \le L_0^r \|y_1 - y_2\|_2, \quad (9)$$
$$\|\nabla r(y_1) - \nabla r(y_2)\|_2 \le L_1^r \|y_1 - y_2\|_2. \quad (10)$$

**Assumption 2.7** (Smoothness of the pre-trained score function). Assume $s_t^{\mathrm{pre}}$ is $L_0^s$-Lipschitz and $L_1^s$-gradient Lipschitz in $y \in \mathbb{R}^d$, i.e., the following holds for any $y_1, y_2 \in \mathbb{R}^d$:

$$\|s_t^{\mathrm{pre}}(y_1) - s_t^{\mathrm{pre}}(y_2)\|_2 \le L_{0,t}^s \|y_1 - y_2\|_2, \quad (11)$$
$$\|\nabla s_t^{\mathrm{pre}}(y_1) - \nabla s_t^{\mathrm{pre}}(y_2)\|_2 \le L_{1,t}^s \|y_1 - y_2\|_2. \quad (12)$$

While Assumptions 2.6 and 2.7 guarantee the smoothness of both the expected reward $r(\cdot)$ and the pre-trained score $\{s_t^{\mathrm{pre}}\}_{t=0}^{T-1}$, no convexity assumptions are imposed, and thus the control problem (2)–(3) is in general *non-concave* with respect to $\{u_t\}_{t=0}^{T-1}$.

Next, we establish the existence and uniqueness of the optimal control, as well as the regularities of the solution to problem (2)–(3). For ease of exposition, we define a series of constants recursively. Let $\lambda_t \in (0, 1)$ for each $t < T$. Set $L_{0,T}^{V^*} = L_0^r$ and $L_{1,T}^{V^*} = L_1^r$. For $t < T$, define

$$L_{0,t}^{V^*} = \frac{1}{\sqrt{\alpha_t}}(1 + (1-\alpha_t)L_{0,t}^s)L_{0,t+1}^{V^*}, \quad (13)$$

$$L_{1,t}^{V^*} = \frac{1}{\alpha_t}(1 + (1-\alpha_t)L_{0,t}^s)(1 + (1-\alpha_t)L_{0,t}^{u^*})L_{1,t+1}^{V^*}$$
$$+ \frac{1-\alpha_t}{\sqrt{\alpha_t}}L_{1,t}^s L_{0,t+1}^{V^*}, \quad (14)$$

$$L_{0,t}^{u^*} = \lambda_t^{-1}\left(L_{0,t}^s + \frac{1-\lambda_t}{1-\alpha_t}\right), \quad (15)$$

$$L_{1,t}^{u^*} = \lambda_t^{-1}\left(L_{1,t}^s + \frac{\mathbb{E}\left[\|W_t\|_2\right](1-\lambda_t)}{(1-\alpha_t)\sqrt{\alpha_t}\sigma_t}\right.$$

$$\left.\times\left(1+(1-\alpha_t)L_{0,t}^{u^*}\right)^2\right). \quad (16)$$

Here, $\mathbb{E}\left[\|W_t\|_2\right] = \frac{\sqrt{2}\Gamma((d+1)/2)}{\Gamma(d/2)} < \infty$ is a constant with $\Gamma(\cdot)$ being the Gamma function. Note that the above constants only depend on the system parameters $\alpha_t$, $L_0^r$, $L_1^r$, $L_0^s$ and $L_1^s$ as well as the hyper-parameter $\lambda_t$. In the next theorem, we show that the above constants are indeed the Lipschitz coefficients for $\{V_t^*\}_{t=0}^T$ and $\{u_t^*\}_{t=0}^{T-1}$.

**Theorem 2.8** (Regularity and well-posedness)**.** *Suppose Assumptions 2.6 and 2.7 hold and let the constants be defined according to (13)–(16). Choose $\beta_t$ such that $1-\frac{\sigma_t^2}{\beta_t}L_{1,t+1}^{V^*} \geq \lambda_t > 0$. Then,*

(i) *The optimal value function $V_t^*(y)$ defined in (7) is $L_{0,t}^{V^*}$-Lipschitz, differentiable and $L_{1,t}^{V^*}$-gradient Lipschitz in $y \in \mathbb{R}^d$.*

(ii) *There is a unique optimal control $u_t^* : \mathbb{R}^d \to \mathbb{R}^d$ of problem (2)–(3) satisfying*

$$u_t^*(y) = s_t^{\mathrm{pre}}(y) + \frac{\sqrt{\alpha_t}\sigma_t^2}{(1-\alpha_t)\beta_t}\mathbb{E}\left[\nabla V_{t+1}^*\left(y'\right)\right], \quad (17)$$

*where $y' = 1/\sqrt{\alpha_t}\left(y+(1-\alpha_t)u_t^*(y)\right) + \sigma_t W_t$. Moreover, the optimal value function $V_t^*$ is the unique $\mathcal{C}^1$ solution to the Bellman equation (8).*

(iii) *The optimal control $u_t^*(y)$ is $L_{0,t}^{u^*}$-Lipschitz, differentiable, and $L_{1,t}^{u^*}$-gradient Lipschitz in $y \in \mathbb{R}^d$.*

When the regularization coefficient $\beta_t$ is sufficiently large, Theorem 2.8 states that the optimal value function $V_t^*(\cdot)$ is Lipschitz and gradient Lipschitz. The choice of $\beta_t$ also ensures the right hand side of (8) is strongly concave in $u_t(\cdot)$, guaranteeing the existence and uniqueness of the optimal control $u_t^*$. Furthermore, Theorem 2.8 shows the optimal control $u_t^*$ is also Lipschitz and gradient Lipschitz. The regularities of $V_t^*$ and $u_t^*$ will serve as the foundation of the algorithm design and convergence analysis in the subsequent sections.

The main challenge in proving Theorem 2.8 stems from the fact that Eq. (17) is an *implicit* equation, in which both sides involve the optimal control $u_t^*$. This prevents a direct application of Lipschitz assumptions on model parameters. We outline a brief proof sketch to highlight the ideas and defer the detailed proof to Appendix B, which is based on backward induction. Assuming $V_{t+1}^*$ is Lipschitz and gradient Lipschitz, the first-order optimality condition implies that (17) holds, given the optimization problem is unconstrained.

Direct calculation leads to the Lipchitz condition of $u_t^*$ and the smoothing effect of the Gaussian random variable is utilized to obtain differentiability. In addition, the Lipschitz property of $\nabla u_t^*$ is a consequence of the integration by parts formula in (35); hence (iii) holds. Finally, the regularities of $V_t^*$ follow from the Lipschitz conditions of $u_t^*$ and $\nabla u_t^*$, which completes the induction.

# 3. Algorithm development and convergence analysis

In this section, we propose an iterative algorithm, PI-FT, for fine-tuning and provide non-asymptotic analysis of its convergence. Assuming that the expected reward function $r(\cdot)$ is known, we develop an iterative algorithm to approximate the optimal policy in high-dimensional settings, suitable for practical implementation. In practice, $r(\cdot)$ is approximated using a small sample set with human feedback, which then serves as the basis for fine-tuning. The unknown $r(\cdot)$ case is left as a topic for future investigation.

## 3.1. Proposed algorithm

Recall that the optimal control $u_t^*$ satisfies (17). Motivated by this observation, we propose the following iterative algorithm for fine-tuning and computing the optimal control $u_t^*$.

---

**Algorithm 1** Policy Iteration for Fine-Tuning (PI-FT)

---

1: **Input:** Expected reward function $r(\cdot)$, pre-trained model $\{s_t^{\mathrm{pre}}\}_{t=0}^T$, and number of iterations $\{m_t\}_{t=0}^{T-1}$.
2: Set $V_T^{(m_T)}(y) = r(y)$ for all $y \in \mathbb{R}^d$.
3: **for** $t = T-1, \ldots, 0$ **do**
4:     Set $u_t^{(0)}(y) = s_t^{\mathrm{pre}}(y)$.
5:     **for** $m = 1, \ldots, m_t - 1$ **do**
6:         Update the control using

$$u_t^{(m+1)}(y) = \frac{\sqrt{\alpha_t}\sigma_t^2}{(1-\alpha_t)\beta_t}\mathbb{E}\left[\nabla V_{t+1}^{(m_{t+1})}\left(y^{(m)}\right)\right]$$
$$+ s_t^{\mathrm{pre}}(y). \quad (18)$$

        where $y^{(m)} = \frac{1}{\sqrt{\alpha_t}}(y+(1-\alpha_t)u_t^{(m)}(y)) + \sigma_t W_t$
7:     **end for**
8:     Compute the value function $V_t^{(m_t)}$ using

$$V_t^{(m_t)}(y) = \mathbb{E}\left[V_{t+1}^{(m_{t+1})}\left(y^{(m_t)}\right)\right.$$
$$\left. - \beta_t\frac{(1-\alpha_t)^2}{2\alpha_t\sigma_t^2}\|u_t^{(m_t)}(y) - s_t^{\mathrm{pre}}(y)\|_2^2\right]. \quad (19)$$

9: **end for**
10: **return** $\left\{u_t^{(m_t)}\right\}_{t=0}^{T-1}$ and $\left\{V_t^{(m_t)}\right\}_{t=0}^T$.

---

Algorithm 1 consists of two nested loops. The inner loop updates the control at each iteration $m$ and the outer loop computes the value function at time $t$. Given the value function $V_{t+1}^{(m_{t+1})}$ at time step $t + 1$, we update $u_t^{(m)}$ using (18) and then evaluate the associated value function using (19). Note that (18) can be viewed as an approximation of the following update rule:

$$u_t^{(m+1)}(y) = s_t^{\text{pre}}(y) + \frac{\sqrt{\alpha_t}\sigma_t^2}{(1 - \alpha_t)\beta_t}\mathbb{E}[\nabla V_{t+1}^*(y^{(m)})], \quad (20)$$

with $y^{(m)} = 1/\sqrt{\alpha_t}\left(y + (1 - \alpha_t)u_t^{(m)}(y)\right) + \sigma_t W_t$, and (19) can be seen as an approximation of the Bellman equation (8), where we replace the unknown $V_{t+1}^*$ with $V_{t+1}^{(m_{t+1})}$. Direct calculation shows that the fixed-point update rule (20) guarantees the convergence of $\left\{u_t^{(m)}\right\}_{m \geq 0}$ to $u_t^*$; see (69). However, in practice we cannot access $V_{t+1}^*$. Hence, Eq. (18) can be viewed as a proxy to (20) as long as we can control the error $\left\|\nabla V_t^{(m_t)}(y) - \nabla V_t^*(y)\right\|_2$ for all $t$. With $u_t^{(m_t)}$ and $V_{t+1}^{(m_{t+1})}$, we use (19) to calculate $V_t^{(m_t)}$, an estimate of the optimal value function $V_t^*$.

### 3.2. Convergence analysis

In this subsection, we analyze the convergence of Algorithm 1. For a fixed $\lambda_t \in (0, 1)$, define a series of constants recursively as follows. Let $L_{0,T}^{\bar{V}} = L_0^r$ and $L_{1,T}^{\bar{V}} = L_1^r$. For each $t < T$, set

$$L_{0,t}^{\bar{V}} = \frac{1}{\sqrt{\alpha_t}}(1 + (1 - \alpha_t)L_{0,t}^s)L_{0,t+1}^{\bar{V}}$$

$$+ \frac{L_{0,t+1}^{\bar{V}}}{\sqrt{\alpha_t}}\left(1 + (1 - \alpha_t)L_{0,t}^{u^*}\right)(1 - \lambda_t), \quad (21)$$

$$L_{1,t}^{\bar{V}} = \frac{1}{\alpha_t}(1 + (1 - \alpha_t)L_{0,t}^s)(1 + (1 - \alpha_t)L_{0,t}^{u^*})L_{1,t+1}^{\bar{V}}$$

$$+ \frac{1 - \alpha_t}{\sqrt{\alpha_t}}L_{1,t}^s L_{0,t+1}^{\bar{V}} + \max_{m \geq 1}\left\{\left(\frac{1 - \alpha_t}{\sqrt{\alpha_t}}L_{1,t}^{u^*} + (m + 2)\right)\right.$$

$$\left. \times \left(1 + (1 - \alpha_t)L_{0,t}^{u^*}\right)^2 \frac{\mathbb{E}\left[\|W_t\|_2\right]}{\alpha_t \sigma_t}\right)L_{0,t+1}^{\bar{V}}(1 - \lambda_t)^{m+1}\right\}. \quad (22)$$

Here, the constant $L_{1,t}^{\bar{V}}$ is well-defined as the maximum must be achieved for some $m < \infty$ due to the exponential decay of $(1 - \lambda_t)^{m+1}$. Similar to the constants defined in (13)–(16), the above constants only depend on the system parameters $\alpha_t$, $L_0^r$, $L_1^r$, $L_0^s$ and $L_1^s$ as well as the hyperparameter $\lambda_t$. Later, we will show that the constants defined in (21) and (22) indeed serve as the Lipschitz coefficients of $\left\{V_t^{(m)}\right\}_{t=0}^T$ universal over all $m \geq 0$. The main result of this subsection is stated as follows.

**Theorem 3.1.** *Suppose Assumptions 2.6 and 2.7 hold. For each $t < T$, choose $\beta_t$ such that $1 - \frac{\sigma_t^2}{\beta_t}L_{1,t+1}^{\bar{V}} \geq \lambda_t > 0$.*

*Let $\left\{u_t^{(m_t)}\right\}_{t=0}^{T-1}$ and $\left\{V_t^{(m_t)}\right\}_{t=0}^T$ be the output of Algorithm 1. Then it holds that*

$$\left\|u_t^{(m_t)}(y) - u_t^*(y)\right\|_2$$

$$\leq \left((1 - \lambda_t)^{m_t}L_{0,t+1}^{V^*} + \lambda_t^{-1}E_{t+1}\right)\frac{\sqrt{\alpha_t}(1 - \lambda_t)}{(1 - \alpha_t)L_{1,t+1}^{V^*}}, \quad (23)$$

*where*

$$E_t := \left\|\nabla V_t^{(m_t)}(y) - \nabla V_t^*(y)\right\|_2$$

$$\leq \sum_{k=t}^{T-1}\left(\prod_{\ell=t}^{k-1}C_{1,\ell}\right)C_{2,k}(1 - \lambda_k)^{m_k+1}, \quad (24)$$

*and*

$$C_{1,t} = \frac{1 + (1 - \alpha_t)L_{0,t}^s}{\lambda_t\sqrt{\alpha_t}}, \quad \text{and}$$

$$C_{2,t} = \frac{\left(1 + (1 - \alpha_t)L_{0,t}^{u^*}\right)}{\sqrt{\alpha_t}}L_{0,t+1}^{\bar{V}}$$

$$+ \frac{\left(1 + (1 - \alpha_t)L_{0,t}^s\right)}{\sqrt{\alpha_t}}L_{0,t+1}^{V^*}.$$

To obtain a simplified convergence rate from Theorem 3.1, we set $\lambda_t \equiv \lambda := \min_{0 \leq t \leq T-1}\lambda_t$ and $m_t \equiv M$ for all $t$. In this case, we have $E_t = \mathcal{O}((1 - \lambda)^{M+1})$ and consequently $\left\|u_t^{(M)}(y) - u_t^*(y)\right\|_2 = \mathcal{O}((1 - \lambda)^M)$. In other words, the control sequence $\left\{u_t^{(m)}\right\}_{t=0}^{T-1}$ converges to the optimal control at a linear rate. Note that the parameter $\lambda$ determines the convergence rate. In particular, setting a larger $\lambda$ results in faster convergence. However, when $\lambda$ is large, Theorem 3.1 requires that the regularization parameter $\beta_t$ must also be chosen to be sufficiently large. Recall that $\beta_t$ controls the distance between the fine-tuned model and the pre-trained model, while a bigger $\lambda$ may result in an optimal control far away from $s_t^{\text{pre}}$. Therefore, choosing an appropriate $\lambda$ (or equivalently $\beta_t$) is crucial to trade-off computational efficiency and closeness to the pre-trained model.

The convergence analysis of the PI-FT algorithm hinges on proving that the regularity of the value function is *preserved* throughout the training process. Here, we present a proof sketch of Theorem 3.1, with the full proof deferred to Appendix C. The key idea is to employ backward induction, breaking the problem down into a sequence of one-step analyses. Assuming $V_{t+1}^{(m_{t+1})}$ is both Lipschitz and gradient Lipschitz, Lemma C.1 establishes the regularities of both $u_t^{(m)}$ and $V_t^{(m)}$ throughout the update rule (for all $m \geq 0$). Furthermore, given the approximation error at time $t + 1$, Lemmas C.2 and C.3 provide the error bounds of both the control and the gradient of value function at time $t$. This completes the induction.

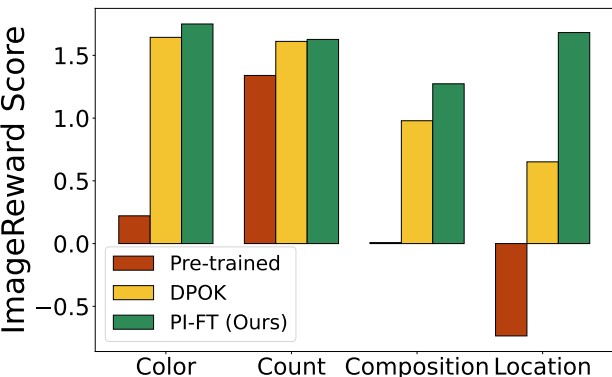

*Figure 1.* Comparison of the ImageReward score among three models: pre-trained Stable Diffusion (red), DPOK (yellow), and PI-FT (green). The ImageReward scores are averaged over 50 samples from each model.

### 3.3. Extension to parametric formulation

To enable practical and efficient update of Algorithm 1, we adopt a linear parameterization in this subsection:

$$u_t(y) = K_t \phi(y), \quad t = 0, \ldots, T-1, \quad (25)$$

where $\phi(y) = (\phi_1(y), \ldots, \phi_p(y))^\top$ is a given basis function, and $\mathbf{K} = \{K_t\}_{t=0}^{T-1}$ is an (unknown) parameter to be learned. Despite its simplicity, this parameterization is flexible, expressive, and tractable, and is widely used in control and reinforcement learning (Agarwal et al., 2021; Jin et al., 2020). With an appropriate choice of basis, it can capture a broad class of score approximators, including random features (Rahimi & Recht, 2007), kernel methods (Steinwart & Christmann, 2008), and overparameterized neural networks in the NTK regime (Jacot et al., 2018). This choice of parameterization enables linear convergence to the optimal solution; details are deferred to Appendix A.

## 4. Numerical Experiments

In this section, we evaluate the performance of the PI-FT algorithm from Section 3 via numerical experiments, focusing on the following questions:

- In practice, how fast does the PI-FT algorithm converge to the optimal solution?
- How does the choice of $\beta$ affect the convergence rate and the quality of the fine-tuned models?

As shown in this section, the PI-FT algorithm converges efficiently to the global optimum; increasing $\beta$ accelerates convergence and yields a model closer to the pre-trained one, aligning with our theoretical analysis in Section 3.

**Model Setup.** We fine-tune the Stable Diffusion v1.5 (Rombach et al., 2022) for text-to-image generation, using

LoRA (Hu et al., 2022) and ImageReward (Xu et al., 2023). Following (Fan et al., 2024), we use four prompts—"A green colored rabbit," "A cat and a dog," "Four wolves in the park," and "A dog on the moon"—to evaluate the model's ability to generate correct color, composition, counting, and location, respectively. During training, we generate 10 trajectories, each consisting of 50 transitions, to calculate the gradient with 1000 gradient steps. By default, we use the AdamW optimizer with a learning rate of $3 \times 10^{-4}$, and set the KL regularization coefficient to a fixed value as $\beta = 0.01$.

**Evaluation.** We first compare ImageReward scores for images generated by the pre-trained model, DPOK (Fan et al., 2024), and our proposed PI-FT. For a fair comparison, we configure DPOK to perform 10 gradient steps per sampling step, using a learning rate of $1 \times 10^{-5}$. Each gradient step is computed using 50 randomly sampled transitions from a replay buffer. As a result, 1000 sampling steps and a total of 10,000 gradient steps in DPOK yield a computational cost comparable to that of PI-FT. As shown in Figure 1, PI-FT consistently outperforms both baselines across all four prompts. Figure 3 further shows that PI-FT more accurately captures object counts and placements (e.g., four wolves and the dog on the moon) and avoids errors like miscoloring the rabbit. It also produces more natural textures compared to the baselines.

**Effect of KL regularization.** KL regularization is known to enhance fine-tuning. We study its effect in PI-FT using the prompt "Four wolves in the park," varying $\beta \in \{0.01, 0.1, 1.0\}$. As shown in Figure 2a, the gradient norm decreases to zero in all cases, indicating convergence. Figure 2b shows that small $\beta$ values improve and stabilize the ImageReward score, while larger $\beta$ offers limited gains. This aligns with Figure 2c, where KL divergence remains high for $\beta = 0.01$, but stays significantly lower for $\beta \in \{0.1, 1.0\}$. Figure 4 also shows that smaller $\beta$ produces images with nearly four wolves, whereas larger $\beta$ leads to fewer. These results underscore the importance of the KL coefficient in effective fine-tuning.

## 5. Conclusion

We introduce a stochastic control framework for fine-tuning diffusion models, integrating linear dynamics with KL regularization. Our approach establishes the well-posedness and regularity of the control problem and proposes a policy iteration algorithm (PI-FT) that guarantees global linear convergence. Unlike prior work that assumes regularity throughout training, we prove that PI-FT inherently maintains these properties. Additionally, we extend our framework to parametric settings, broadening its applicability. This work advances the theoretical understanding of fine-tuning diffusion models and provides a foundation for developing more effective fine-tuning algorithms. Our algorithmic design and theoretical findings are also supported by thorough numerical experiments.

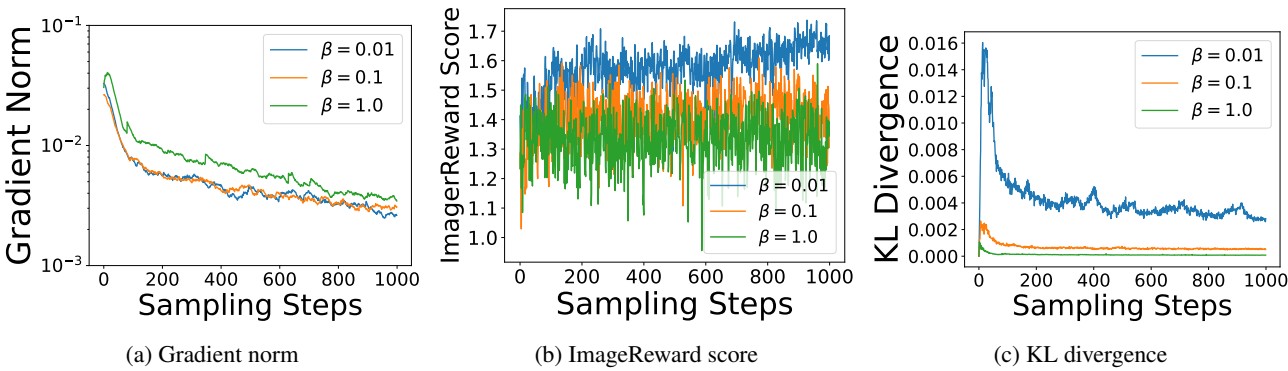

(a) Gradient norm      (b) ImageReward score      (c) KL divergence

*Figure 2.* (a) Gradient norm (logarithmic scale) of PI-FT during training. The curves are smoothed using the exponential moving average (EMA). A linear convergence rate is observed. (b) ImageReward score of PI-FT during training. Smaller KL regularization coefficient $\beta$ leads to higher ImageReward score. (c) KL divergence of PI-FT during training. Larger KL regularization coefficient $\beta$ leads to smaller KL divergence and faster convergence.

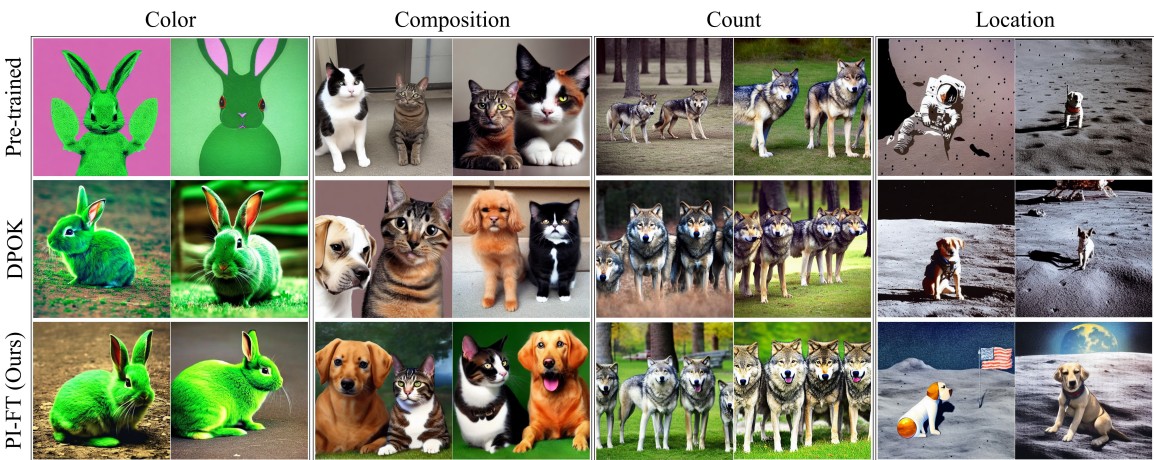

*Figure 3.* Visual comparison of images generated by the original Stable Diffusion model (pre-trained), DPOK model, and PI-FT model (ours). Prompts from left to right: "A green colored rabbit" (color), "A cat and a dog" (composition), "Four wolves in the park" (count), and "A dog on the moon" (location).

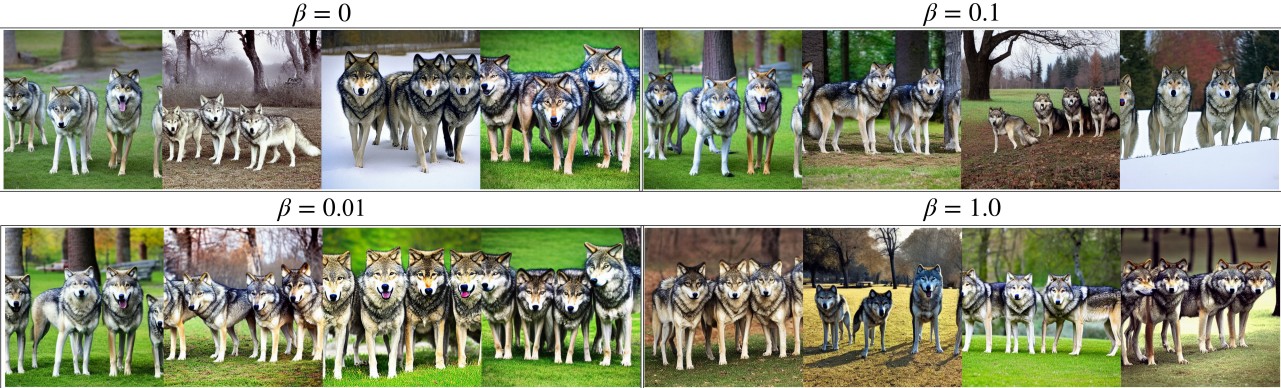

*Figure 4.* Randomly generated samples from PI-FT model with different KL regularization coefficients. Images from a single text prompt: "Four wolves in the park". The model with smaller $\beta > 0$ generates more accurate number of wolves.

## Impact statement

This paper presents a technical contribution in the form of a fine-tuning algorithm. We do not identify any specific societal impacts that warrant mention.

## Acknowledgements

R.X. is partially supported by the NSF CAREER award DMS-2339240 and a JP Morgan Research Award. The work of M.R. is partially supported by Google Research and Meta.

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

# A. Extension to parametric formulation

In this section, we provide a roadmap to show how a linear convergence rate can be achieved with the linear parameterization (25). We make the following realizability assumption.

**Assumption A.1** (Realizability). Assume that at each timestamp $t$, the optimal control satisfies $u_t^*(y) = K_t^* \phi(y)$ for some matrix $K_t^* \in \mathbb{R}^{p \times d}$.

Define the value function associated with the policy $\mathbf{K}$ as

$$J(\mathbf{K}) := \mathbb{E}_{Y_0 \sim \mathcal{D}} \left[ V_0^{\mathbf{K}}(Y_0) \right],$$

where $\mathcal{D}$ is the distribution of the initial state and we can take $\mathcal{D} = \mathcal{N}(0, I_d)$, which is the distribution of $Y_0^{\text{pre}}$. Here the value function is defined as

$$V_t^{\mathbf{K}}(y) := \mathbb{E}\left[ -\sum_{\ell=t}^{T-1} \beta_\ell \frac{(1-\alpha_\ell)^2}{2\alpha_\ell \sigma_\ell^2} \|K_\ell \phi(Y_\ell) - s_\ell^{\text{pre}}(Y_\ell)\|_2^2 + R(Y_T) \Big| Y_t = y \right],$$

with terminal condition

$$V_T^{\mathbf{K}}(y) = r(y) = \mathbb{E}\left[ R(y) \right].$$

Similarly, define the Q-function:

$$Q_t^{\mathbf{K}}(y, u) := \mathbb{E}\left[ V_{t+1}^{\mathbf{K}} \left( \frac{1}{\sqrt{\alpha_t}} (y + (1-\alpha_t)u) + \sigma_t W_t \right) - \beta_t \frac{(1-\alpha_t)^2}{2\alpha_t \sigma_t^2} \|u - s_t(y)\|_2^2 \right],$$

where the expecation is taken over $W_t$. We remark that the policy $\mathbf{K}$ in the superscript refers to $\{K_\ell\}_{\ell > t}$ at each $t$. Recall the choice of $\beta_t$ implies the map $u \mapsto Q_t^{\mathbf{K}^*}(y, u)$ is strongly concave uniformly over all $y$; see Theorem 2.8. If the feature mapping $\phi$ is well-behaved, the map $K_t \mapsto Q_t^{\mathbf{K}^*}(y, K_t \phi(y))$ is also strongly concave.

Next, we calculate the policy gradient with respect to $\mathbf{K}$. Note that

$$\partial_t J(\mathbf{K}) := \frac{\partial J(\mathbf{K})}{\partial K_t} = \frac{\partial}{\partial K_t} \mathbb{E}\left[ -\sum_{\ell=0}^{t-1} \beta_\ell \frac{(1-\alpha_\ell)^2}{2\alpha_\ell \sigma_\ell^2} \|K_\ell \phi(Y_\ell) - s_\ell^{\text{pre}}(Y_\ell)\|_2^2 + Q_t^{\mathbf{K}}(Y_t, K_t \phi(Y_t)) \right]$$

$$= \frac{\partial}{\partial K_t} \mathbb{E}\left[ Q_t^{\mathbf{K}}(Y_t, K_t \phi(Y_t)) \right], \tag{26}$$

where the expectation is taken over the initial state $Y_0 \sim \mathcal{D}$ and noise $\{W_t\}_{t=0}^{T-1}$. We next show that the optimal policy $\mathbf{K}^*$ is the unique stationary point of $J(\mathbf{K})$. Let $\overline{\mathbf{K}}$ be a stationary point, i.e., $\partial_t J(\mathbf{K})\big|_{\mathbf{K}=\overline{\mathbf{K}}} = 0$ for all $t$. In particular, we have

$$\frac{\partial}{\partial K_{T-1}} \mathbb{E}\left[ Q_{T-1}^{\mathbf{K}}(Y_{T-1}, K_{T-1}\phi(Y_{T-1})) \right] \Big|_{\mathbf{K}=\overline{\mathbf{K}}} = 0.$$

The strong concavity in $K_{T-1}$ implies that $K_{T-1}^* = \overline{K}_{T-1}$. Similarly, we have

$$\frac{\partial}{\partial K_{T-2}} \mathbb{E}\left[ Q_{T-2}^{\mathbf{K}}(Y_{T-2}, K_{T-2}\phi(Y_{T-2})) \right] \Big|_{\mathbf{K}=\overline{\mathbf{K}}} = \frac{\partial}{\partial K_{T-2}} \mathbb{E}\left[ Q_{T-2}^{\mathbf{K}^*}(Y_{T-2}, K_{T-2}\phi(Y_{T-2})) \right] \Big|_{\mathbf{K}=\overline{\mathbf{K}}} = 0.$$

By strong concavity again, we have $K_{T-2}^* = \overline{K}_{T-2}$. Repeating the argument to conclude that $\mathbf{K}^* = \overline{\mathbf{K}}$ is the unique stationary point of the objective function $J(\mathbf{K})$. We remark that the above analysis relies on the two facts:

(i) At $T-1$, the map $K_{T-1} \mapsto \mathbb{E}\left[ Q_{T-1}^{\mathbf{K}}(Y_{T-1}, K_{T-1}\phi(Y_{T-1})) \right]$ has a unique stationary point.

(ii) For any $t < T-1$, the map $K_t \mapsto \mathbb{E}\left[ Q_t^{\mathbf{K}^*}(Y_t, K_t \phi(Y_t)) \right]$ has a unique stationary point.

These two conditions (i)–(ii) *relax* Bhandari & Russo (2024, Assumption 2.A), which assumes that $K_t \mapsto \mathbb{E}\left[Q_t^{\mathbf{K}}(Y_t, K_t\phi(Y_t))\right]$ has no sub-optimal stationary points under *any* $\mathbf{K}$. While (Bhandari & Russo, 2024) primarily focuses on global convergence guarantees for infinite-horizon problems, it also introduces a condition (*cf.* Condition 4) for finite-horizon MDPs that aligns in spirit with our analysis.

Notably, Bhandari & Russo (2024) *did not* analyze the convergence rate of policy gradient methods for finite-horizon problems. Since $\mathbf{K}^*$ is the unique stationary point, the policy gradient method converges to the globally optimal solution as long as the standard smoothness condition is satisfied. Moreover, we conjecture a *linear convergence rate* due to the presence of strong concavity. Consider the policy gradient update rule:

$$K_t^{(m+1)} = K_t^{(m)} + \eta\partial_t J(\mathbf{K}^{(m)}), \;\; 0 \le t \le T-1, m \ge 0.$$

For each $t < T$, we have

$$
\begin{aligned}
\left\|K_t^{(m+1)} - K_t^*\right\|_F &= \left\|K_t^{(m)} + \eta\partial_t J(\mathbf{K}^{(m)}) - K_t^*\right\|_F \\
&\le \left\|K_t^{(m)} + \eta\partial_t J(\mathbf{K}_{\le t}^{(m)}, \mathbf{K}_{>t}^*) - K_t^*\right\|_F + \eta\left\|\partial_t J(\mathbf{K}_{\le t}^{(m)}, \mathbf{K}_{>t}^*) - \partial_t J(\mathbf{K}^{(m)})\right\|_F,
\end{aligned}
$$

where $\mathbf{K}_{\le t} \coloneqq \{K_\ell\}_{\ell=0}^t$ and $\mathbf{K}_{>t} \coloneqq \{K_\ell\}_{\ell>t}$. Eq. (26) implies $\partial_t J(\mathbf{K}) = \partial_t J(\mathbf{K}_{\ge t})$ is independent of $\{K_\ell\}_{\ell<t}$. The (one-step) strong concavity and smoothness leads to

$$\left\|K_t^{(m)} + \eta\partial_t J(\mathbf{K}_{\le t}^{(m)}, \mathbf{K}_{>t}^*) - K_t^*\right\|_F \le c\left\|K_t^{(m)} - K_t^*\right\|_F,$$

for some constant $c < 1$. If $\partial_t J(\mathbf{K})$ is further $L_{1,t}^J$-Lipschitz in $\mathbf{K} \in \mathbb{R}^{p\times d}$, then we have

$$\left\|\partial_t J(\mathbf{K}_{\le t}^{(m)}, \mathbf{K}_{>t}^*) - \partial_t J(\mathbf{K}^{(m)})\right\|_F \le L_{1,t}^J\left\|\mathbf{K}_{>t}^{(m)} - \mathbf{K}_{>t}^*\right\|_F,$$

and consequently we prove the linear convergence rate of the policy gradient method.

# B. Ommitted proofs in Section 2

## B.1. Proof of Lemma 2.5

*Proof.* Recall that for any $y_t \in \mathbb{R}^d$,

$$p_{t+1|t}(\cdot|y_t) = f\left(\cdot|\mu_t(y_t), \sigma_t^2 I_d\right) \;\; \text{and} \;\; p_{t+1|t}^{\text{pre}}(\cdot|y_t) = f\left(\cdot|\mu_t^{\text{pre}}(y_t), \sigma_t^2 I_d\right),$$

with

$$\mu_t(y_t) = \frac{1}{\sqrt{\alpha_t}}\left(y_t + (1-\alpha_t)u_t(y_t)\right) \;\; \text{and} \;\; \mu_t^{\text{pre}}(y_t) = \frac{1}{\sqrt{\alpha_t}}\left(y_t + (1-\alpha_t)s_t^{\text{pre}}(y_t)\right),$$

where $f(\cdot|\mu, \Sigma)$ is the Gaussian density with mean $\mu \in \mathbb{R}^d$ and covariance $\Sigma \in \mathbb{R}^{d\times d}$. Thus, for any $y_t, y_{t+1} \in \mathbb{R}^d$, we have

$$\log\left(\frac{p_{t+1|t}(y_{t+1}|y_t)}{p_{t+1|t}^{\text{pre}}(y_{t+1}|y_t)}\right) = -\frac{1}{2\sigma_t^2}\|y_{t+1} - \mu_t(y_t)\|_2^2 + \frac{1}{2\sigma_t^2}\|y_{t+1} - \mu_t^{\text{pre}}(y_t)\|_2^2.$$

Denote $\mathbb{E}_{p_{t+1|t}}$ as the expectation under the conditional density $p_{t+1|t}(\cdot|y_t)$ of $Y_{t+1}$ given $Y_t = y_t$. By definition of the KL divergence, we have

$$
\begin{aligned}
\text{KL}\left(p_{t+1|t}(\cdot|y_t)\|p_{t+1|t}^{\text{pre}}(\cdot|y_t)\right) &= \mathbb{E}_{p_{t+1|t}}\left[\log\left(\frac{p_{t+1|t}(Y_{t+1}|y_t)}{p_{t+1|t}^{\text{pre}}(Y_{t+1}|y_t)}\right)\right] \\
&= -\frac{1}{2\sigma_t^2}\mathbb{E}_{p_{t+1|t}}\left[\|Y_{t+1} - \mu_t(y_t)\|_2^2\right] + \frac{1}{2\sigma_t^2}\mathbb{E}_{p_{t+1|t}}\left[\|Y_{t+1} - \mu_t^{\text{pre}}(y_t)\|_2^2\right]. \quad (27)
\end{aligned}
$$

Note that

$$
\mathbb{E}_{p_{t+1|t}} \left[ \|Y_{t+1} - \mu_t^{\mathrm{pre}}(y_t)\|_2^2 \right] = \mathbb{E}_{p_{t+1|t}} \left[ \|Y_{t+1} - \mu_t(y_t)\|_2^2 \right] + \mathbb{E}_{p_{t+1|t}} \left[ \|\mu_t(y_t) - \mu_t^{\mathrm{pre}}(y_t)\|_2^2 \right]
$$
$$
+ 2\mathbb{E}_{p_{t+1|t}} \left[ (Y_{t+1} - \mu_t(y_t))^\top (\mu_t(y_t) - \mu_t^{\mathrm{pre}}(y_t)) \right]
$$
$$
= \mathbb{E}_{p_{t+1|t}} \left[ \|Y_{t+1} - \mu_t(y_t)\|_2^2 \right] + \|\mu_t(y_t) - \mu_t^{\mathrm{pre}}(y_t)\|_2^2,
$$

where we use the fact that $\mathbb{E}_{t+1|t}[Y_{t+1}] = \mu_t(y_t)$. Plugging the above equality into (27), we obtain

$$
\mathrm{KL}\Big(p_{t+1|t}(\,\cdot\,|y_t) \| p_{t+1|t}^{\mathrm{pre}}(\,\cdot\,|y_t)\Big) = \frac{1}{2\sigma_t^2} \|\mu_t(y_t) - \mu_t^{\mathrm{pre}}(y_t)\|_2^2 = \frac{(1-\alpha_t)^2}{2\alpha_t\sigma_t^2} \|u_t(y_t) - s_t^{\mathrm{pre}}(y_t)\|_2^2,
$$

which completes the proof. $\qquad\square$

### B.2. Proof of Theorem 2.8

*Proof.* We prove Theorem 2.8 by backward induction. At time $t = T$, Assumption 2.6 implies $V_T^*(y) = r(y)$ is $L_{0,T}^{V^*}$-Lipschitz and $L_{1,T}^{V^*}$-gradient Lipschitz with $L_{0,T}^{V^*} = L_0^r$ and $L_{1,T}^{V^*} = L_1^r$.

Step 1: Lipschitz condition of $u_t^*$. Assume that $V_{t+1}^*$ is $L_{0,t+1}^{V^*}$-Lipschitz and $L_{1,t+1}^{V^*}$-gradient Lipschitz in $y \in \mathbb{R}^d$. The choice of $\beta_t$ implies that the mapping

$$
u \mapsto \mathbb{E}\left[ V_{t+1}^*\left( \frac{1}{\sqrt{\alpha_t}}(y + (1-\alpha_t)u) + \sigma_t W_t \right) - \beta_t \frac{(1-\alpha_t)^2}{2\alpha_t\sigma_t^2} \|u - s_t^{\mathrm{pre}}(y)\|_2^2 \right] \tag{28}
$$

is $\gamma_t$-strongly concave with $\gamma_t = \frac{(1-\alpha_t)^2}{\alpha_t}\left( \frac{\beta_t}{\sigma_t^2} - L_{1,t+1}^{V^*} \right) > 0$ for any $y \in \mathbb{R}^d$. Hence, there is a unique optimal control $u_t^*$ satisfying

$$
u_t^*(y) = s_t^{\mathrm{pre}}(y) + \frac{\sqrt{\alpha_t}\sigma_t^2}{(1-\alpha_t)\beta_t} \mathbb{E}\left[ \nabla V_{t+1}^*\left( \frac{1}{\sqrt{\alpha_t}}(y + (1-\alpha_t)u_t^*(y)) + \sigma_t W_t \right) \right], \tag{29}
$$

which is obtained by setting the gradient of mapping (28) as zero for each $y \in \mathbb{R}^d$. Here, we apply the Lipschitz condition of $V_{t+1}^*$ and the dominated convergence theorem to interchange the operators.

We next prove the Lipschitz condition of $u_t^*$ in $y \in \mathbb{R}^d$. Note that for any $y_1$ and $y_2$ in $\mathbb{R}^d$, Eq. (29) implies

$$
u_t^*(y_1) - u_t^*(y_2) = s_t^{\mathrm{pre}}(y_1) - s_t^{\mathrm{pre}}(y_2) + \frac{\sqrt{\alpha_t}\sigma_t^2}{(1-\alpha_t)\beta_t}\mathbb{E}\left[ \nabla V_{t+1}^*\left( \frac{1}{\sqrt{\alpha_t}}(y_1 + (1-\alpha_t)u_t^*(y_1)) + \sigma_t W_t \right) \right]
$$
$$
- \frac{\sqrt{\alpha_t}\sigma_t^2}{(1-\alpha_t)\beta_t}\mathbb{E}\left[ \nabla V_{t+1}^*\left( \frac{1}{\sqrt{\alpha_t}}(y_2 + (1-\alpha_t)u_t^*(y_2)) + \sigma_t W_t \right) \right].
$$

Utilizing the Lipschitz condition of $s_t^{\mathrm{pre}}$ and $\nabla V_{t+1}^*$, we obtain

$$
\|u_t^*(y_1) - u_t^*(y_2)\|_2 \leq \|s_t^{\mathrm{pre}}(y_1) - s_t^{\mathrm{pre}}(y_2)\|_2 + \frac{\sqrt{\alpha_t}\sigma_t^2}{(1-\alpha_t)\beta_t}\mathbb{E}\Big[\Big\| \nabla V_{t+1}^*\left( \frac{1}{\sqrt{\alpha_t}}(y_1 + (1-\alpha_t)u_t^*(y_1)) + \sigma_t W_t \right)
$$
$$
- \nabla V_{t+1}^*\left( \frac{1}{\sqrt{\alpha_t}}(y_2 + (1-\alpha_t)u_t^*(y_2)) + \sigma_t W_t \right) \Big\|_2\Big]
$$
$$
\leq L_{0,t}^s \|y_1 - y_2\|_2
$$
$$
+ \frac{\sqrt{\alpha_t}\sigma_t^2}{(1-\alpha_t)\beta_t}L_{1,t+1}^{V^*} \left\| \frac{1}{\sqrt{\alpha_t}}(y_1 + (1-\alpha_t)u_t^*(y_1)) - \frac{1}{\sqrt{\alpha_t}}(y_2 + (1-\alpha_t)u_t^*(y_2)) \right\|_2
$$
$$
\leq L_{0,t}^s \|y_1 - y_2\|_2 + \frac{\sqrt{\alpha_t}\sigma_t^2}{(1-\alpha_t)\beta_t}L_{1,t+1}^{V^*}\left( \frac{1}{\sqrt{\alpha_t}}\|y_1 - y_2\|_2 + \frac{1-\alpha_t}{\sqrt{\alpha_t}}\|u_t^*(y_1) - u_t^*(y_2)\|_2 \right).
$$

Equivalently, we have

$$
\left( 1 - \frac{\sigma_t^2}{\beta_t}L_{1,t+1}^{V^*} \right)\|u_t^*(y_1) - u_t^*(y_2)\|_2 \leq \left( L_{0,t}^s + \frac{\sigma_t^2 L_{1,t+1}^{V^*}}{(1-\alpha_t)\beta_t} \right)\|y_1 - y_2\|_2.
$$

Since $1 - \frac{\sigma_t^2}{\beta_t} L_{1,t+1}^{V^*} \geq \lambda_t > 0$, we established the Lipschitz condition of the optimal control $u_t^*$:

$$\|u_t^*(y_1) - u_t^*(y_2)\|_2 \leq \lambda_t^{-1} \left( L_{0,t}^s + \frac{1 - \lambda_t}{1 - \alpha_t} \right) \|y_1 - y_2\|_2 = L_{0,t}^{u^*} \|y_1 - y_2\|_2 \,.,$$

where we recall $L_{0,t}^{u^*}$ defined in (15).

Step 2: Differentiability of $u_t^*$. We now argue that $u_t^*(y)$ is differentiable for all $y \in \mathbb{R}^d$. Let $h \in \mathbb{R}^d$ be an arbitrary non-zero vector and let $y \in \mathbb{R}^d$ be fixed. Since $s_t^{\text{pre}}$ is differentiable, we have

$$s_t^{\text{pre}}(y + h) - s_t^{\text{pre}}(y) = \nabla s_t^{\text{pre}}(y)h + o(\|h\|_2). \tag{30}$$

Moreover, by the inductive hypothesis, $\nabla V_{t+1}^*(y)$ is Lipschitz and thus $\nabla^2 V_{t+1}^*(y)$ exists for almost all $y$ and is bounded. Folland (1999, Theorem 2.27) implies the mapping $z \mapsto \mathbb{E}\left[ \nabla V_{t+1}^*(z + \sigma_t W_t) \right]$ is differentiable everywhere and its derivative is given by

$$\frac{\partial}{\partial z} \mathbb{E}\left[ \nabla V_{t+1}^*(z + \sigma_t W_t) \right] = \mathbb{E}\left[ \nabla^2 V_{t+1}^*(z + \sigma_t W_t) \right]. \tag{31}$$

Let $z = \frac{1}{\sqrt{\alpha_t}} (y + (1 - \alpha_t) u_t^*(y))$ and $k = \frac{1}{\sqrt{\alpha_t}} (h + (1 - \alpha_t)(u_t^*(y + h) - u_t^*(y))$. Eq. (31) implies

$$\mathbb{E}\left[ \nabla V_{t+1}^*(z + k + \sigma_t W_t) \right] - \mathbb{E}\left[ \nabla V_{t+1}^*(z + \sigma_t W_t) \right] = \mathbb{E}\left[ \nabla^2 V_{t+1}^*(z + \sigma_t W_t) \right] k + o(\|k\|_2). \tag{32}$$

Since $u_t^*$ is Lipschitz, we have $k = \mathcal{O}(\|h\|_2)$ as $\|h\|_2 \to 0$. Combining (30) and (32), we obtain

$$u_t^*(y + h) - u_t^*(y) = s_t^{\text{pre}}(y + h) - s_t^{\text{pre}}(y) + \frac{\sqrt{\alpha_t}\sigma_t^2}{(1 - \alpha_t)\beta_t} \mathbb{E}\left[ \nabla V_{t+1}^* \left( \frac{1}{\sqrt{\alpha_t}} ((y + h) + (1 - \alpha_t) u_t^*(y + h)) + \sigma_t W_t \right) \right]$$

$$- \frac{\sqrt{\alpha_t}\sigma_t^2}{(1 - \alpha_t)\beta_t} \mathbb{E}\left[ \nabla V_{t+1}^* \left( \frac{1}{\sqrt{\alpha_t}} (y + (1 - \alpha_t) u_t^*(y)) + \sigma_t W_t \right) \right]$$

$$= \nabla s_t^{\text{pre}}(y)h + \frac{\sqrt{\alpha_t}\sigma_t^2}{(1 - \alpha_t)\beta_t} \mathcal{H}_{t+1}(y) \left( \frac{1}{\sqrt{\alpha_t}} (h + (1 - \alpha_t)(u_t^*(y + h) - u_t^*(y))) \right) + o(\|h\|_2),$$

where $\mathcal{H}_{t+1}(y) := \mathbb{E}\left[ \nabla^2 V_{t+1}^* \left( \frac{1}{\sqrt{\alpha_t}} (y + (1 - \alpha_t) u_t^*(y)) + \sigma_t W_t \right) \right]$. Re-arranging the terms leads to

$$u_t^*(y + h) - u_t^*(y) = \left( I_d - \frac{\sigma_t^2}{\beta_t} \mathcal{H}_{t+1}(y) \right)^{-1} \left( \nabla s_t^{\text{pre}}(y) + \frac{\sigma_t^2}{(1 - \alpha_t)\beta_t} \mathcal{H}_{t+1}(y) \right) h + o(\|h\|_2),$$

which proves that $u_t^*(y)$ is differentiable for any $y \in \mathbb{R}^d$ and its derivative is given by

$$\nabla u_t^*(y) = \left( I_d - \frac{\sigma_t^2}{\beta_t} \mathcal{H}_{t+1}(y) \right)^{-1} \left( \nabla s_t^{\text{pre}}(y) + \frac{\sigma_t^2}{(1 - \alpha_t)\beta_t} \mathcal{H}_{t+1}(y) \right). \tag{33}$$

Step 3: Lipschitz condition of $\nabla u_t^*$. Next, we show that $\nabla u_t^*$ is Lipschitz. Taking the derivative of both sides of (29),

$$\nabla u_t^*(y) = \nabla s_t^{\text{pre}}(y) + \frac{\sqrt{\alpha_t}\sigma_t^2}{(1 - \alpha_t)\beta_t} \cdot \frac{\partial}{\partial y} \mathbb{E}\left[ \nabla V_{t+1}^* \left( \frac{1}{\sqrt{\alpha_t}} (y + (1 - \alpha_t) u_t^*(y)) + \sigma_t W_t \right) \right]. \tag{34}$$

For any $z \in \mathbb{R}^d$, the integration by parts formula implies that

$$\frac{\partial}{\partial z} \mathbb{E}\left[ \nabla V_{t+1}^*(z + \sigma_t W_t) \right] = \mathbb{E}\left[ \nabla^2 V_{t+1}^*(z + \sigma_t W_t) \right] = \mathbb{E}\left[ \nabla V_{t+1}^*(z + \sigma_t W_t) \frac{W_t^\top}{\sigma_t} \right]. \tag{35}$$

Substituting $z = \frac{1}{\sqrt{\alpha_t}} (y + (1 - \alpha_t) u_t^*(y))$ and applying the chain rule, we obtain

$$\frac{\partial}{\partial y} \mathbb{E}\left[ \nabla V_{t+1}^* \left( \frac{1}{\sqrt{\alpha_t}} (y + (1 - \alpha_t) u_t^*(y)) + \sigma_t W_t \right) \right] = \mathcal{W}_{t+1}(y)\mathcal{U}_t(y),$$

where $\mathcal{W}_{t+1}(y) := \mathbb{E}\left[\nabla V_{t+1}^*\left(\frac{1}{\sqrt{\alpha_t}}(y+(1-\alpha_t)u_t^*(y))+\sigma_t W_t\right)\frac{W_t^\top}{\sigma_t}\right]$ and $\mathcal{U}_t(y) := \frac{1}{\sqrt{\alpha_t}}(I_d+(1-\alpha_t)\nabla u_t^*(y))$. It follows from (34) that

$$\nabla u_t^*(y) = \nabla s_t^{\mathrm{pre}}(y) + \frac{\sqrt{\alpha_t}\sigma_t^2}{(1-\alpha_t)\beta_t}\mathcal{W}_{t+1}(y)\mathcal{U}_t(y). \tag{36}$$

For any $y_1$ and $y_2$ in $\mathbb{R}^d$, Eq. (36) implies that

$$\nabla u_t^*(y_1) - \nabla u_t^*(y_2) = \nabla s_t^{\mathrm{pre}}(y_1) - \nabla s_t^{\mathrm{pre}}(y_2) + \frac{\sqrt{\alpha_t}\sigma_t^2}{(1-\alpha_t)\beta_t}\left(\mathcal{W}_{t+1}(y_1)\mathcal{U}_t(y_1) - \mathcal{W}_{t+1}(y_2)\mathcal{U}_t(y_2)\right). \tag{37}$$

Note that for any $y \in \mathbb{R}^d$ we have

$$\|\mathcal{W}_{t+1}(y)\|_2 = \|\mathcal{H}_{t+1}(y)\|_2 \le L_{1,t+1}^{V^*}, \tag{38}$$

$$\|\mathcal{U}_t(y)\|_2 \le \frac{1}{\sqrt{\alpha_t}}\left(1+(1-\alpha_t)L_{0,t}^{u^*}\right), \tag{39}$$

where (38) holds by applying the identity (35), and (39) is a consequence of the Lipschitz condition of $u_t^*$. Moreover, for any $y_1$ and $y_2$ in $\mathbb{R}^d$, the Lipschitz conditions of $\nabla V_{t+1}^*$ and $u_t^*$ imply that

$$\|\mathcal{W}_{t+1}(y_1) - \mathcal{W}_{t+1}(y_2)\|_2 \le \frac{\mathbb{E}\left[\|W_t\|_2\right]}{\sigma_t}L_{1,t+1}^{V^*}\frac{1}{\sqrt{\alpha_t}}\left(\|y_1-y_2\|_2+(1-\alpha_t)L_{0,t}^{u^*}\|y_1-y_2\|_2\right)$$

$$= \frac{\mathbb{E}\left[\|W_t\|_2\right]}{\sqrt{\alpha_t}\sigma_t}L_{1,t+1}^{V^*}\left(1+(1-\alpha_t)L_{0,t}^{u^*}\right)\|y_1-y_2\|_2. \tag{40}$$

Furthermore, the definition of $\mathcal{U}_t$ implies

$$\|\mathcal{U}_t(y_1) - \mathcal{U}_t(y_2)\|_2 = \frac{1-\alpha_t}{\sqrt{\alpha_t}}\|\nabla u_t^*(y_1) - \nabla u_t^*(y_2)\|_2. \tag{41}$$

Combining (37)–(41) together, we have

$$\|\nabla u_t^*(y_1) - \nabla u_t^*(y_2)\|_2 \le \|\nabla s_t^{\mathrm{pre}}(y_1) - \nabla s_t^{\mathrm{pre}}(y_2)\|_2 + \frac{\sqrt{\alpha_t}\sigma_t^2}{(1-\alpha_t)\beta_t}\|\mathcal{W}_{t+1}(y_1) - \mathcal{W}_{t+1}(y_2)\|_2\|\mathcal{U}_t(y_1)\|_2$$

$$+ \frac{\sqrt{\alpha_t}\sigma_t^2}{(1-\alpha_t)\beta_t}\|\mathcal{W}_{t+1}(y_2)\|_2\|\mathcal{U}_t(y_1) - \mathcal{U}_t(y_2)\|_2$$

$$\le L_{1,t}^s\|y_1-y_2\|_2 + \frac{\sqrt{\alpha_t}\sigma_t^2}{(1-\alpha_t)\beta_t}\left(L_{1,t+1}^{V^*}\frac{1-\alpha_t}{\sqrt{\alpha_t}}\|\nabla u_t^*(y_1) - \nabla u_t^*(y_2)\|_2\right.$$

$$\left.+ \frac{1}{\alpha_t}\left(1+(1-\alpha_t)L_{0,t}^{u^*}\right)^2\frac{\mathbb{E}\left[\|W_t\|_2\right]}{\sigma_t}L_{1,t+1}^{V^*}\|y_1-y_2\|_2\right).$$

Since $\frac{\sigma_t^2}{\beta_t}L_{1,t+1}^{V^*} \le 1-\lambda_t$, we deduce that

$$\|\nabla u_t^*(y_1) - \nabla u_t^*(y_2)\|_2 \le \lambda_t^{-1}\left(L_{1,t}^s + \frac{\mathbb{E}\left[\|W_t\|_2\right](1-\lambda_t)}{(1-\alpha_t)\sqrt{\alpha_t}\sigma_t}\left(1+(1-\alpha_t)L_{0,t}^{u^*}\right)^2\right)\|y_1-y_2\|_2$$

$$= L_{1,t}^{u^*}\|y_1-y_2\|_2,$$

where we recall $L_{1,t}^{u^*}$ defined in (16).

Step 4: Lipschitz conditions of $V_t^*$ and $\nabla V_t^*$. Finally, we turn to prove the Lipschitz and gradient Lipschitz conditions of $V_t^*$. Plugging $u_t^*$ into the Bellman equation (8), we have

$$V_t^*(y) = \mathbb{E}\left[V_{t+1}^*\left(\frac{1}{\sqrt{\alpha_t}}(y+(1-\alpha_t)u_t^*(y))+\sigma_t W_t\right)\right] - \beta_t\frac{(1-\alpha_t)^2}{2\alpha_t\sigma_t^2}\|u_t^*(y)-s_t^{\mathrm{pre}}(y)\|_2^2. \tag{42}$$

Since $V_{t+1}^*$ and $s_t^{\text{pre}}$ are differentiable with Lipschitz gradients and $u_t^*$ is differentiable, we know that $V_t^*$ is differentiable and

$$\nabla V_t^*(y) = \frac{\partial}{\partial y} \mathbb{E}\left[V_{t+1}^*\left(\frac{1}{\sqrt{\alpha_t}}(y + (1-\alpha_t)u_t^*(y)) + \sigma_t W_t\right) - \beta_t \frac{(1-\alpha_t)^2}{2\alpha_t \sigma_t^2}\|u_t^*(y) - s_t^{\text{pre}}(y)\|_2^2\right]$$

$$= \frac{1}{\sqrt{\alpha_t}}(I_d + (1-\alpha_t)\nabla u_t^*(y))^\top \mathbb{E}\left[\nabla V_{t+1}^*\left(\frac{1}{\sqrt{\alpha_t}}(y + (1-\alpha_t)u_t^*(y)) + \sigma_t W_t\right)\right]$$

$$- \beta_t \frac{(1-\alpha_t)^2}{\alpha_t \sigma_t^2}(\nabla u_t^*(y) - \nabla s_t^{\text{pre}}(y))^\top (u_t^* - s_t^{\text{pre}}(y)).$$

Define $\mathcal{G}_{t+1}(y) := \mathbb{E}\left[\nabla V_{t+1}^*\left(\frac{1}{\sqrt{\alpha_t}}(y + (1-\alpha_t)u_t^*(y)) + \sigma_t W_t\right)\right]$. It follows that

$$
\begin{aligned}
\nabla V_t^*(y) &= \frac{1}{\sqrt{\alpha_t}}(I_d + (1-\alpha_t)\nabla u_t^*(y))^\top \mathcal{G}_{t+1}(y) - \beta_t \frac{(1-\alpha_t)^2}{\alpha_t \sigma_t^2}(\nabla u_t^*(y) - \nabla s_t^{\text{pre}}(y))^\top \left(\frac{\sqrt{\alpha_t}\sigma_t^2}{(1-\alpha_t)\beta_t}\mathcal{G}_{t+1}(y)\right) \\
&= \frac{1}{\sqrt{\alpha_t}}(I_d + (1-\alpha_t)\nabla s_t^{\text{pre}}(y))^\top \mathcal{G}_{t+1}(y),
\end{aligned}
\tag{43}
$$

where (29) is applied to obtain the first equality. Next, we use (43) to establish the Lipschitz conditions of $V_t^*$ and $\nabla V_t^*$. The Lipschitz condition of $\nabla V_{t+1}^*$ implies $\|\mathcal{G}_{t+1}(y)\|_2 \leq L_{0,t+1}^{V^*}$ for all $y \in \mathbb{R}^d$. Based on (43), we have

$$\|\nabla V_t^*(y)\|_2 \leq \frac{1}{\sqrt{\alpha_t}}(1 + (1-\alpha_t)L_{0,t}^s)L_{0,t+1}^{V^*} = L_{0,t}^{V^*},$$

which proves the Lipschitz condition of $V_t^*$. Next, we prove the gradient Lipschitz condition of $V_t^*$. For ease of exposition, we denote $\mathcal{S}_t(y) := \frac{1}{\sqrt{\alpha_t}}(I_d + (1-\alpha_t)\nabla s_t^{\text{pre}}(y))$. We now use (43) to show the Lipschitz condition of $\nabla V_t^*$. We first note that $\nabla \mathcal{G}_{t+1}(y) = \mathcal{H}_{t+1}(y)\mathcal{U}_t(y)$ is well-defined at every $y \in \mathbb{R}^d$, and thus

$$\|\nabla \mathcal{G}_{t+1}(y)\|_2 \leq \|\mathcal{H}_{t+1}(y)\|_2 \|\mathcal{U}_t(y)\|_2 \leq \frac{1}{\sqrt{\alpha_t}}(1 + (1-\alpha_t)L_{0,t}^{u^*})L_{1,t+1}^{V^*}.$$

With the Lipschitz condition of $\mathcal{G}_{t+1}$ in hand, for any $y_1$ and $y_2$ in $\mathbb{R}^d$, we have that

$$
\begin{aligned}
\|\nabla V_t^*(y_1) - \nabla V_t^*(y_2)\|_2 &\leq \left\|\mathcal{S}_t(y_1)^\top \mathcal{G}_{t+1}(y_1) - \mathcal{S}_t(y_2)^\top \mathcal{G}_{t+1}(y_2)\right\|_2 \\
&\leq \left\|\mathcal{S}_t(y_1)^\top (\mathcal{G}_{t+1}(y_1) - \mathcal{G}_{t+1}(y_2))\right\|_2 + \left\|(\mathcal{S}_t(y_1) - \mathcal{S}_t(y_2))^\top \mathcal{G}_{t+1}(y_2)\right\|_2 \\
&\leq \|\mathcal{S}_t(y_1)\|_2 \|\mathcal{G}_{t+1}(y_1) - \mathcal{G}_{t+1}(y_2)\|_2 + \|\mathcal{S}_t(y_1) - \mathcal{S}_t(y_2)\|_2 \|\mathcal{G}_{t+1}(y_2)\|_2 \\
&\leq \frac{1}{\alpha_t}(1 + (1-\alpha_t)L_{0,t}^s)(1 + (1-\alpha_t)L_{0,t}^{u^*})L_{1,t+1}^{V^*}\|y_1 - y_2\|_2 \\
&\quad + \frac{1-\alpha_t}{\sqrt{\alpha_t}}L_{1,t}^s L_{0,t+1}^{V^*}\|y_1 - y_2\|_2,
\end{aligned}
$$

where the last equation uses the Lipschitz conditions of $s_t^{\text{pre}}$, $\nabla s_t^{\text{pre}}$, $V_{t+1}^*$ and $\nabla V_{t+1}^*$. Consequently, we have

$$\|\nabla V_t^*(y_1) - \nabla V_t^*(y_2)\|_2 \leq L_{1,t}^{V^*}\|y_1 - y_2\|_2,$$

where we recall $L_{1,t}^{V^*}$ defined in (14). In other words, $L_{1,t}^{V^*}$ is indeed the Lipschitz constant of $\nabla V_t^*$. This completes the proof.

$\square$

## C. Omitted proofs in Section 3

To prove Theorem 3.1, we need a few intermediate results for one-step update rule. We begin with a lemma that characterizes the regularity of control and value functions after one update. Given a function $\widehat{V}_{t+1} : \mathbb{R}^d \to \mathbb{R}$, define the following update rule:

$$u_t^{(m+1)}(y) = s_t^{\text{pre}}(y) + \frac{\sqrt{\alpha_t}\sigma_t^2}{(1-\alpha_t)\beta_t}\mathbb{E}\left[\nabla \widehat{V}_{t+1}\left(\frac{1}{\sqrt{\alpha_t}}\left(y + (1-\alpha_t)u_t^{(m)}(y)\right) + \sigma_t W_t\right)\right], \tag{44}$$

with initialization $u_t^{(0)}(y) = s_t^{\mathrm{pre}}(y)$. Furthermore, let $V_t^{(m)}$ be the value function induced by $u_t^{(m)}$, i.e.,

$$V_t^{(m)}(y) = \mathbb{E}\left[\widehat{V}_{t+1}\left(\frac{1}{\sqrt{\alpha_t}}\left(y + (1-\alpha_t)u_t^{(m)}(y)\right) + \sigma_t W_t\right) - \beta_t \frac{(1-\alpha_t)^2}{2\alpha_t \sigma_t^2}\|u_t^{(m)}(y) - s_t^{\mathrm{pre}}(y)\|^2\right]. \quad (45)$$

Later, we will choose $\widehat{V}_{t+1} = V_{t+1}^{(m_{t+1})}$ to analyze the convergence of Algorithm 1. With (44) and (45), we state the lemma as follows.

**Lemma C.1** (One-step regularity and universal upper bound). *Suppose Assumptions 2.6 and 2.7 hold. Let $\widehat{V}_{t+1}$ be a function that is $L_{0,t+1}^{\widehat{V}}$-Lipschitz and $L_{1,t+1}^{\widehat{V}}$-gradient Lipschitz. Consider $\left\{u_t^{(m)}\right\}_{m=0}^{m_t}$ and $\left\{V_t^{(m)}\right\}_{m=0}^{m_t}$ defined in (44) and (45). For $t < T$, choose $\beta_t$ such that $1 - \frac{\sigma_t^2}{\beta_t}L_{1,t+1}^{\widehat{V}} \geq \lambda_t > 0$. Then it holds for every $m \geq 0$ that*

*(i) $u_t^{(m)}$ is $L_{0,t}^{u^{(m)}}$-Lipschitz and $L_{1,t}^{u^{(m)}}$-gradient Lipschitz, with coefficients $L_{0,t}^{u^{(m)}}$ and $L_{1,t}^{u^{(m)}}$ satisfying*

$$L_{0,t}^{u^{(m)}} \leq L_{0,t}^{u^*} \quad \text{and} \quad L_{1,t}^{u^{(m)}} \leq L_{1,t}^{u^*}. \quad (46)$$

*(ii) $V_t^{(m)}$ is $L_{0,t}^{V^{(m)}}$-Lipschitz and $L_{1,t}^{V^{(m)}}$-gradient Lipschitz, with coefficients $L_{0,t}^{V^{(m)}}$ and $L_{1,t}^{V^{(m)}}$ satisfying*

$$L_{0,t}^{V^{(m)}} \leq \frac{1}{\sqrt{\alpha_t}}(1 + (1-\alpha_t)L_{0,t}^s)L_{0,t+1}^{\widehat{V}} + \frac{L_{0,t+1}^{\widehat{V}}}{\sqrt{\alpha_t}}\left(1 + (1-\alpha_t)L_{0,t}^{u^*}\right)(1-\lambda_t)^{m+1}, \quad (47)$$

$$\begin{aligned}
L_{1,t}^{V^{(m)}} &\leq \frac{1}{\alpha_t}(1 + (1-\alpha_t)L_{0,t}^s)(1 + (1-\alpha_t)L_{0,t}^{u^*})L_{1,t+1}^{\widehat{V}} + \frac{1-\alpha_t}{\sqrt{\alpha_t}}L_{1,t}^s L_{0,t+1}^{\widehat{V}} \\
&\quad + \left(\frac{1-\alpha_t}{\sqrt{\alpha_t}}L_{1,t}^{u^*} + (m+2)\left(1 + (1-\alpha_t)L_{0,t}^{u^*}\right)^2 \frac{\mathbb{E}\left[\|W_t\|_2\right]}{\alpha_t \sigma_t}\right)L_{0,t+1}^{\widehat{V}}(1-\lambda_t)^{m+1}. \quad (48)
\end{aligned}$$

Lemma C.1 provides regularity properties of the sequence of controls and value functions generated by (44) and (45) throughout the optimization process. Specifically, the Lipschitz and gradient Lipschitz constants of $u_t^{(m)}$ are bounded by those of the optimal control $u_t^*$; see (46). Furthermore, when $\widehat{V}_{t+1} = V_{t+1}^*$, Eqs. (47) and (48) lead to

$$L_{0,t}^{V^{(m)}} = L_{0,t}^{V^*} + \mathcal{O}((1-\lambda_t)^{m+1}) \quad \text{and} \quad L_{1,t}^{V^{(m)}} = L_{1,t}^{V^*} + \mathcal{O}((m+2)(1-\lambda_t)^{m+1}).$$

The residual terms diminish to zero as $m \to \infty$ and thus providing the convergence of Lipschitz constants. Consequently, selecting $\widehat{V}_{t+1}$ as $V_{t+1}^{(m_{t+1})}$, when feasible, directly ensures the regularity of the control and value functions defined in Algorithm 1.

We outline a brief proof sketch to highlight the ideas before providing the detailed proof. The Lipschitz condition of $u_t^{(m)}$ is derived through direct calculation, while the gradient Lipschitz condition leverages the integration by parts formula (50). Additionally, establishing the regularity of $V_t^{(m)}$ requires a careful analysis of the gradient expression, along with a tightly controlled upper bound; see (57).

*Proof of Lemma C.1.* Step 1: Lipschitz conditions of $u_t^{(m)}$ and $\nabla u_t^{(m)}$. We begin with the Lipschitz condition of $u_t^{(m)}$. For any $y_1$ and $y_2$ in $\mathbb{R}^d$, the update rule (44) implies

$$\begin{aligned}
\left\|u_t^{(m+1)}(y_1) - u_t^{(m+1)}(y_2)\right\|_2 &\leq L_{0,t}^s \|y_1 - y_2\|_2 \\
&\quad + \frac{\sqrt{\alpha_t}\sigma_t^2}{(1-\alpha_t)\beta_t}L_{1,t+1}^{\widehat{V}}\left(\frac{1}{\sqrt{\alpha_t}}\|y_1 - y_2\|_2 + \frac{1-\alpha_t}{\sqrt{\alpha_t}}\left\|u_t^{(m)}(y_1) - u_t^{(m)}(y_2)\right\|_2\right).
\end{aligned}$$

Since $1 - \frac{\sigma_t^2}{\beta_t}L_{1,t+1}^{\widehat{V}} \geq \lambda_t$, unrolling the recursion leads to

$$L_{0,t}^{u^{(m+1)}} \leq L_{0,t}^s + \frac{\sigma_t^2}{(1-\alpha_t)\beta_t}L_{1,t+1}^{\widehat{V}} + \frac{\sigma_t^2}{\beta_t}L_{1,t+1}^{\widehat{V}}L_{0,t}^{u^{(m)}}$$

$$\leq \lambda_t^{-1}\left(L_{0,t}^s + \frac{1-\lambda_t}{1-\alpha_t}\right) = L_{0,t}^{u^*}.$$

Here, we also use the fact that $L_{0,t}^{u^{(0)}} = L_{0,t}^s$. Note that the condition $1 - \frac{\sigma_t^2}{\beta_t}L_{1,t+1}^{\widehat{V}} \geq \lambda_t$ is important as it decouples the upper bound, in the sense that the last line does not depend on $L_{1,t+1}^{\widehat{V}}$ directly.

Next, since $u_t^{(0)} = s_t^{\mathrm{pre}}$ is differentiable, a simple inductive argument shows that $\nabla u_t^{(m)}$ is well-defined. Furthermore, we show that $\nabla u_t^{(m)}$ is Lipschitz. Differentiate both sides of the update rule (44),

$$\nabla u_t^{(m+1)}(y) = \nabla s_t^{\mathrm{pre}}(y) + \frac{\sqrt{\alpha_t}\sigma_t^2}{(1-\alpha_t)\beta_t} \cdot \frac{\partial}{\partial y}\mathbb{E}\left[\nabla\widehat{V}_{t+1}\left(\frac{1}{\sqrt{\alpha_t}}\left(y + (1-\alpha_t)u_t^{(m)}(y)\right) + \sigma_t W_t\right)\right]. \tag{49}$$

For any $z \in \mathbb{R}^d$, the integration by parts formula implies that

$$\frac{\partial}{\partial z}\mathbb{E}\left[\nabla\widehat{V}_{t+1}\left(z + \sigma_t W_t\right)\right] = \mathbb{E}\left[\nabla^2\widehat{V}_{t+1}\left(z + \sigma_t W_t\right)\right] = \mathbb{E}\left[\nabla\widehat{V}_{t+1}\left(z + \sigma_t W_t\right)\frac{W_t^\top}{\sigma_t}\right]. \tag{50}$$

Substituting $z = \frac{1}{\sqrt{\alpha_t}}\left(y + (1-\alpha_t)u_t^{(m)}(y)\right)$ and applying the chain rule, we have

$$\frac{\partial}{\partial y}\mathbb{E}\left[\nabla\widehat{V}_{t+1}\left(\frac{1}{\sqrt{\alpha_t}}\left(y + (1-\alpha_t)u_t^{(m)}(y)\right) + \sigma_t W_t\right)\right]$$
$$= \mathbb{E}\left[\nabla\widehat{V}_{t+1}\left(\frac{1}{\sqrt{\alpha_t}}\left(y + (1-\alpha_t)u_t^{(m)}(y)\right) + \sigma_t W_t\right)\frac{W_t^\top}{\sigma_t}\right]\left(\frac{1}{\sqrt{\alpha_t}}\left(I_d + (1-\alpha_t)\nabla u_t^{(m)}(y)\right)\right). \tag{51}$$

Define

$$\widehat{\mathcal{G}}_{t+1}^{(m)}(y) := \mathbb{E}\left[\nabla\widehat{V}_{t+1}\left(\frac{1}{\sqrt{\alpha_t}}\left(y + (1-\alpha_t)u_t^{(m)}(y)\right) + \sigma_t W_t\right)\right],$$

$$\widehat{\mathcal{W}}_{t+1}^{(m)}(y) := \mathbb{E}\left[\nabla\widehat{V}_{t+1}\left(\frac{1}{\sqrt{\alpha_t}}\left(y + (1-\alpha_t)u_t^{(m)}(y)\right) + \sigma_t W_t\right)\frac{W_t^\top}{\sigma_t}\right],$$

$$\mathcal{U}_t^{(m)}(y) := \frac{1}{\sqrt{\alpha_t}}\left(I_d + (1-\alpha_t)\nabla u_t^{(m)}(y)\right).$$

It follows from (51) that $\frac{\partial}{\partial y}\widehat{\mathcal{G}}_{t+1}^{(m)}(y) = \widehat{\mathcal{W}}_{t+1}^{(m)}(y)\mathcal{U}_t^{(m)}(y)$. Consequently, Eq. (49) becomes

$$\nabla u_t^{(m+1)}(y) = \nabla s_t^{\mathrm{pre}}(y) + \frac{\sqrt{\alpha_t}\sigma_t^2}{(1-\alpha_t)\beta_t}\widehat{\mathcal{W}}_{t+1}^{(m)}(y)\mathcal{U}_t^{(m)}(y). \tag{52}$$

Since $\nabla\widehat{V}_{t+1}$ and $u_t^{(m)}$ are both Lipschitz, we know that $\nabla u_t^{(m+1)}$ is Lipschitz as long as $\nabla u_t^{(m)}$ is Lipschitz. Specifically, for any $y_1$ and $y_2$ in $\mathbb{R}^d$, we have

$$\nabla u_t^{(m+1)}(y_1) - \nabla u_t^{(m+1)}(y_2) = \nabla s_t^{\mathrm{pre}}(y_1) - \nabla s_t^{\mathrm{pre}}(y_2) + \frac{\sqrt{\alpha_t}\sigma_t^2}{(1-\alpha_t)\beta_t}\left(\widehat{\mathcal{W}}_{t+1}^{(m)}(y_1)\mathcal{U}_t^{(m)}(y_1) - \widehat{\mathcal{W}}_{t+1}^{(m)}(y_2)\mathcal{U}_t^{(m)}(y_2)\right).$$

Note that for any $y \in \mathbb{R}^d$ we have

$$\left\|\widehat{\mathcal{W}}_{t+1}^{(m)}(y)\right\|_2 \leq L_{1,t+1}^{\widehat{V}}, \tag{53}$$

$$\left\|\mathcal{U}_t^{(m)}(y)\right\|_2 \leq \frac{1}{\sqrt{\alpha_t}}\left(1 + (1-\alpha_t)L_{0,t}^{u^*}\right), \tag{54}$$

where the (53) holds by applying the identity (50), and (54) is a consequence of the Lipschitz condition of $u_t^{(m)}$. Moreover, for any $y_1$ and $y_2$ in $\mathbb{R}^d$, the Lipschitz conditions of $\nabla\widehat{V}_{t+1}$ and $u_t^{(m)}$ imply that

$$\left\|\widehat{\mathcal{W}}_{t+1}^{(m)}(y_1) - \widehat{\mathcal{W}}_{t+1}^{(m)}(y_2)\right\|_2 \leq \frac{\mathbb{E}\left[\|W_t\|_2\right]}{\sigma_t}L_{1,t+1}^{\widehat{V}}\frac{1}{\sqrt{\alpha_t}}\left(\|y_1 - y_2\|_2 + (1-\alpha_t)L_{0,t}^{u^{(m)}}\|y_1 - y_2\|_2\right)$$

$$= \frac{\mathbb{E}\left[\|W_t\|_2\right]}{\sqrt{\alpha_t}\sigma_t}L_{1,t+1}^{\widehat{V}}\left(1+(1-\alpha_t)L_{0,t}^{u^*}\right)\|y_1-y_2\|_2, \tag{55}$$

and the Lipschitz condition of $\nabla u_t^{(m)}$ leads to

$$\left\|\mathcal{U}_t^{(m)}(y_1)-\mathcal{U}_t^{(m)}(y_2)\right\|_2 = \frac{1-\alpha_t}{\sqrt{\alpha_t}}\left\|\nabla u_t^{(m)}(y_1)-\nabla u_t^{(m)}(y_2)\right\|_2$$

$$\leq \frac{1-\alpha_t}{\sqrt{\alpha_t}}L_{1,t}^{u^{(m)}}\|y_1-y_2\|_2. \tag{56}$$

Combine (52)–(56) together to have

$$\left\|\nabla u_t^{(m+1)}(y_1)-\nabla u_t^{(m+1)}(y_2)\right\|_2 \leq L_{1,t}^s\|y_1-y_2\|_2 + \frac{\sqrt{\alpha_t}\sigma_t^2}{(1-\alpha_t)\beta_t}\left(L_{1,t+1}^{\widehat{V}}\frac{1-\alpha_t}{\sqrt{\alpha_t}}L_{1,t}^{u^{(m)}}\right.$$

$$\left. + \frac{1}{\alpha_t}\left(1+(1-\alpha_t)L_{0,t}^{u^*}\right)^2\frac{\mathbb{E}\left[\|W_t\|_2\right]}{\sigma_t}L_{1,t+1}^{\widehat{V}}\right)\|y_1-y_2\|_2,$$

Since $\frac{\sigma_t^2}{\beta_t}L_{1,t+1}^{\widehat{V}} \leq 1-\lambda_t$, we deduce that

$$L_{1,t}^{u^{(m+1)}} \leq L_{1,t}^s + \left(L_{1,t}^{u^{(m)}} + \frac{\mathbb{E}\left[\|W_t\|_2\right]}{(1-\alpha_t)\sqrt{\alpha_t}\sigma_t}\left(1+(1-\alpha_t)L_{0,t}^{u^*}\right)^2\right)(1-\lambda_t).$$

Equivalently, we have

$$L_{1,t}^{u^{(m)}} \leq \lambda_t^{-1}\left(L_{1,t}^s + \frac{\mathbb{E}\left[\|W_t\|_2\right](1-\lambda_t)}{(1-\alpha_t)\sqrt{\alpha_t}\sigma_t}\left(1+(1-\alpha_t)L_{0,t}^{u^*}\right)^2\right) = L_{1,t}^{u^*}.$$

Step 2: Lipschitz conditions of $V_t^{(m)}$ and $\nabla V_t^{(m)}$. With the Lipschitz conditions of $u_t^{(m)}$ and $\nabla u_t^{(m)}$, we now turn to establish the regularity of $V_t^{(m)}$. Note that the expressions in (44), (45) and (52) imply that

$$\nabla V_t^{(m)}(y)$$

$$= \frac{1}{\sqrt{\alpha_t}}\left(I_d + (1-\alpha_t)\nabla u_t^{(m)}(y)\right)^\top\widehat{\mathcal{G}}_{t+1}^{(m)}(y) - \beta_t\frac{(1-\alpha_t)^2}{\alpha_t\sigma_t^2}(\nabla u_t^{(m)}(y)-\nabla s_t^{\mathrm{pre}}(y))^\top\left(\frac{\sqrt{\alpha_t}\sigma_t^2}{(1-\alpha_t)\beta_t}\widehat{\mathcal{G}}_{t+1}^{(m-1)}(y)\right)$$

$$= \frac{1}{\sqrt{\alpha_t}}\left(I_d + (1-\alpha_t)\nabla s_t^{\mathrm{pre}}(y)\right)^\top\widehat{\mathcal{G}}_{t+1}^{(m)}(y) + \frac{(1-\alpha_t)}{\sqrt{\alpha_t}}\left(\nabla u_t^{(m)}(y)-\nabla s_t^{\mathrm{pre}}(y)\right)^\top\left(\widehat{\mathcal{G}}_{t+1}^{(m)}(y)-\widehat{\mathcal{G}}_{t+1}^{(m-1)}(y)\right). \tag{57}$$

We will use the above expression (57) to derive the Lipschitz and gradient Lipschitz conditions of $V_t^{(m)}$. To proceed, we first get a few useful estimates. Recall the notation $\mathcal{S}_t(y) = \frac{1}{\sqrt{\alpha_t}}\left(I_d + (1-\alpha_t)\nabla s_t^{\mathrm{pre}}(y)\right)$. We first show that $\mathcal{S}_t(y)^\top\widehat{\mathcal{G}}_{t+1}^{(m)}(y)$ is bounded and Lipschitz. Note that for any $y \in \mathbb{R}^d$,

$$\left\|\mathcal{S}_t(y)^\top\widehat{\mathcal{G}}_{t+1}^{(m)}(y)\right\|_2 \leq \|\mathcal{S}_t(y)\|_2\left\|\widehat{\mathcal{G}}_{t+1}^{(m)}(y)\right\|_2 \leq \frac{1}{\sqrt{\alpha_t}}(1+(1-\alpha_t)L_{0,t}^s)L_{0,t+1}^{\widehat{V}}. \tag{58}$$

Also, for any $y_1$ and $y_2$ in $\mathbb{R}^d$, we have

$$\left\|\mathcal{S}_t(y_1)^\top\widehat{\mathcal{G}}_{t+1}^{(m)}(y_1)-\mathcal{S}_t(y_2)^\top\widehat{\mathcal{G}}_{t+1}^{(m)}(y_2)\right\|_2 \leq \left\|\mathcal{S}_t(y_1)^\top\left(\widehat{\mathcal{G}}_{t+1}^{(m)}(y_1)-\widehat{\mathcal{G}}_{t+1}^{(m)}(y_2)\right)\right\|_2 + \left\|(\mathcal{S}_t(y_1)-\mathcal{S}_t(y_2))^\top\widehat{\mathcal{G}}_{t+1}^{(m)}(y_2)\right\|_2$$

$$\leq \|\mathcal{S}_t(y_1)\|_2\left\|\widehat{\mathcal{G}}_{t+1}^{(m)}(y_1)-\widehat{\mathcal{G}}_{t+1}^{(m)}(y_2)\right\|_2 + \|\mathcal{S}_t(y_1)-\mathcal{S}_t(y_2)\|_2\left\|\widehat{\mathcal{G}}_{t+1}^{(m)}(y_2)\right\|_2$$

$$\leq \frac{1}{\alpha_t}(1+(1-\alpha_t)L_{0,t}^s)(1+(1-\alpha_t)L_{0,t}^{u^*})L_{1,t+1}^{\widehat{V}}\|y_1-y_2\|_2$$

$$+ \frac{1-\alpha_t}{\sqrt{\alpha_t}}L_{1,t}^sL_{0,t+1}^{\widehat{V}}\|y_1-y_2\|_2, \tag{59}$$

where we apply the Lipschitz conditions of $\widehat{V}_{t+1}$, $\nabla\widehat{V}_{t+1}$, and $s_t^{\mathrm{pre}}$ in the assumptions and the Lipschitz condition of $u_t^{(m)}$ in Step 1. It remains to show the second term in (57) is bounded and Lipschitz uniformly over $m \geq 0$. Since $\nabla\widehat{V}_{t+1}$ is $L_{1,t+1}^{\widehat{V}}$-Lipschitz, we deduce that

$$
\begin{aligned}
\left\|\widehat{\mathcal{G}}_{t+1}^{(m)}(y) - \widehat{\mathcal{G}}_{t+1}^{(m-1)}(y)\right\|_2 &\leq L_{1,t+1}^{\widehat{V}} \frac{1-\alpha_t}{\sqrt{\alpha_t}} \left\|u_t^{(m)}(y) - u_t^{(m-1)}(y)\right\|_2 \\
&\leq L_{1,t+1}^{\widehat{V}} \frac{\sigma_t^2}{\beta_t} \left\|\widehat{\mathcal{G}}_{t+1}^{(m-1)}(y) - \widehat{\mathcal{G}}_{t+1}^{(m-2)}(y)\right\|_2,
\end{aligned}
\tag{60}
$$

where we apply the update rule (44) in the last equation. As $\frac{\sigma_t^2}{\beta_t} L_{1,t+1}^{\widehat{V}} \leq 1 - \lambda_t$, unrolling the recursion (60) leads to

$$
\begin{aligned}
\left\|\widehat{\mathcal{G}}_{t+1}^{(m)}(y) - \widehat{\mathcal{G}}_{t+1}^{(m-1)}(y)\right\|_2 &\leq (1-\lambda_t)^{m-1} \left\|\widehat{\mathcal{G}}_{t+1}^{(1)}(y) - \widehat{\mathcal{G}}_{t+1}^{(0)}(y)\right\|_2 \\
&\leq (1-\lambda_t)^{m-1} L_{1,t+1}^{\widehat{V}} \frac{1-\alpha_t}{\sqrt{\alpha_t}} \left\|u_t^{(1)} - u_t^{(0)}\right\|_2 \\
&\leq (1-\lambda_t)^{m-1} L_{1,t+1}^{\widehat{V}} \frac{\sigma_t^2}{\beta_t} L_{0,t+1}^{\widehat{V}} \leq (1-\lambda_t)^m L_{0,t+1}^{\widehat{V}}.
\end{aligned}
\tag{61}
$$

Moreover, for any $y \in \mathbb{R}^d$, we have

$$
\begin{aligned}
\left\|\frac{\partial}{\partial y}\left(\widehat{\mathcal{G}}_{t+1}^{(m)}(y) - \widehat{\mathcal{G}}_{t+1}^{(m-1)}(y)\right)\right\|_2 &= \left\|\widehat{\mathcal{W}}_{t+1}^{(m)}(y)\mathcal{U}_t^{(m)}(y) - \widehat{\mathcal{W}}_{t+1}^{(m-1)}(y)\mathcal{U}_t^{(m-1)}(y)\right\|_2 \\
&\leq \left\|\widehat{\mathcal{W}}_{t+1}^{(m)}(y) - \widehat{\mathcal{W}}_{t+1}^{(m-1)}(y)\right\|_2 \left\|\mathcal{U}_t^{(m)}(y)\right\|_2 + \left\|\widehat{\mathcal{W}}_{t+1}^{(m-1)}(y)\right\|_2 \left\|\mathcal{U}_t^{(m)}(y) - \mathcal{U}_t^{(m-1)}(y)\right\|_2.
\end{aligned}
\tag{62}
$$

The gradient expression (52) implies

$$
\left\|\mathcal{U}_t^{(m)}(y) - \mathcal{U}_t^{(m-1)}(y)\right\|_2 \leq \frac{1-\alpha_t}{\sqrt{\alpha_t}} \left\|\nabla u_t^{(m)}(y) - \nabla u_t^{(m-1)}(y)\right\|_2 \leq \frac{\sigma_t^2}{\beta_t} \left\|\frac{\partial}{\partial y}\left(\widehat{\mathcal{G}}_{t+1}^{(m-1)}(y) - \widehat{\mathcal{G}}_{t+1}^{(m-2)}(y)\right)\right\|_2, \tag{63}
$$

and (60)–(61) lead to the fact that

$$
\left\|\widehat{\mathcal{W}}_{t+1}^{(m)}(y) - \widehat{\mathcal{W}}_{t+1}^{(m-1)}(y)\right\|_2 \leq \frac{\mathbb{E}\left[\|W_t\|_2\right]}{\sigma_t} L_{1,t+1}^{\widehat{V}} \frac{1-\alpha_t}{\sqrt{\alpha_t}} \left\|u_t^{(m)}(y) - u_t^{(m-1)}(y)\right\|_2 \leq \frac{\mathbb{E}\left[\|W_t\|_2\right]}{\sigma_t} L_{0,t+1}^{\widehat{V}}(1-\lambda_t)^m. \tag{64}
$$

With (53), (54), (63) and (64), we can bound (62) with

$$
\begin{aligned}
\left\|\frac{\partial}{\partial y}\left(\widehat{\mathcal{G}}_{t+1}^{(m)}(y) - \widehat{\mathcal{G}}_{t+1}^{(m-1)}(y)\right)\right\|_2 &\leq \frac{L_{0,t+1}^{\widehat{V}}\mathbb{E}\left[\|W_t\|_2\right]}{\sqrt{\alpha_t}\sigma_t}\left(1 + (1-\alpha_t)L_{0,t}^{u^*}\right)(1-\lambda_t)^m \\
&\quad + \frac{\sigma_t^2}{\beta_t} L_{1,t+1}^{\widehat{V}} \left\|\frac{\partial}{\partial y}\left(\widehat{\mathcal{G}}_{t+1}^{(m-1)}(y) - \widehat{\mathcal{G}}_{t+1}^{(m-2)}(y)\right)\right\|_2.
\end{aligned}
\tag{65}
$$

Since $\frac{\sigma_t^2}{\beta_t} L_{1,t+1}^{\widehat{V}} \leq 1 - \lambda_t$, unrolling the recursion (65) to have

$$
\begin{aligned}
\left\|\frac{\partial}{\partial y}\left(\widehat{\mathcal{G}}_{t+1}^{(m)}(y) - \widehat{\mathcal{G}}_{t+1}^{(m-1)}(y)\right)\right\|_2 &\leq \frac{L_{0,t+1}^{\widehat{V}}\mathbb{E}\left[\|W_t\|_2\right]}{\sqrt{\alpha_t}\sigma_t}\left(1 + (1-\alpha_t)L_{0,t}^{u^*}\right)m(1-\lambda_t)^m \\
&\quad + (1-\lambda_t)^{(m)} \left\|\frac{\partial}{\partial y}\widehat{\mathcal{G}}_{t+1}^{(0)}(y)\right\|_2 \\
&\leq \frac{L_{0,t+1}^{\widehat{V}}\mathbb{E}\left[\|W_t\|_2\right]}{\sqrt{\alpha_t}\sigma_t}\left(1 + (1-\alpha_t)L_{0,t}^{u^*}\right)(m+1)(1-\lambda_t)^m,
\end{aligned}
\tag{66}
$$

where we utilize the fact that

$$
\left\|\frac{\partial}{\partial y}\widehat{\mathcal{G}}_{t+1}^{(0)}(y)\right\|_2 \leq \left\|\widehat{\mathcal{W}}_{t+1}^{(0)}(y)\right\|_2 \left\|\mathcal{U}_t^{(0)}(y)\right\|_2 \leq \frac{L_{0,t+1}^{\widehat{V}}\mathbb{E}\left[\|W_t\|_2\right]}{\sqrt{\alpha_t}\sigma_t}\left(1 + (1-\alpha_t)L_{0,t}^{u^*}\right).
$$

Hence, for any $y_1$ and $y_2$ in $\mathbb{R}^d$, (53)–(56), (61) and (66) imply that

$$
\begin{aligned}
&\left\| \left( \widehat{\mathcal{W}}_{t+1}^{(m)}(y_1) \mathcal{U}_t^{(m)}(y_1) \right)^\top \left( \widehat{\mathcal{G}}_{t+1}^{(m)}(y_1) - \widehat{\mathcal{G}}_{t+1}^{(m-1)}(y_1) \right) - \left( \widehat{\mathcal{W}}_{t+1}^{(m)}(y_2) \mathcal{U}_t^{(m)}(y_2) \right)^\top \left( \widehat{\mathcal{G}}_{t+1}^{(m)}(y_2) - \widehat{\mathcal{G}}_{t+1}^{(m-1)}(y_2) \right) \right\|_2 \\
&\leq \left\| \widehat{\mathcal{W}}_{t+1}^{(m)}(y_1) \mathcal{U}_t^{(m)}(y_1) - \widehat{\mathcal{W}}_{t+1}^{(m)}(y_2) \mathcal{U}_t^{(m)}(y_2) \right\|_2 \left\| \widehat{\mathcal{G}}_{t+1}^{(m)}(y_1) - \widehat{\mathcal{G}}_{t+1}^{(m-1)}(y_1) \right\|_2 \\
&\quad + \left\| \widehat{\mathcal{W}}_{t+1}^{(m)}(y_2) \mathcal{U}_t^{(m)}(y_2) \right\|_2 \sup_{y \in \mathbb{R}^d} \left\| \frac{\partial}{\partial y} \left( \widehat{\mathcal{G}}_{t+1}^{(m)}(y) - \widehat{\mathcal{G}}_{t+1}^{(m-1)}(y) \right) \right\|_2 \|y_1 - y_2\|_2 \\
&\leq \left( \frac{1-\alpha_t}{\sqrt{\alpha_t}} L_{1,t}^{u^{(m)}} + \left( 1 + (1-\alpha_t) L_{0,t}^{u^*} \right)^2 \frac{\mathbb{E}\left[ \|W_t\|_2 \right]}{\alpha_t \sigma_t} \right) L_{1,t+1}^{\widehat{V}} L_{0,t+1}^{\widehat{V}} (1-\lambda_t)^m \|y_1 - y_2\|_2 \\
&\quad + \left( 1 + (1-\alpha_t) L_{0,t}^{u^*} \right)^2 \frac{\mathbb{E}\left[ \|W_t\|_2 \right]}{\alpha_t \sigma_t} L_{1,t+1}^{\widehat{V}} L_{0,t+1}^{\widehat{V}} (m+1)(1-\lambda_t)^m \|y_1 - y_2\|_2 \\
&= \left( \frac{1-\alpha_t}{\sqrt{\alpha_t}} L_{1,t}^{u^*} + (m+2) \left( 1 + (1-\alpha_t) L_{0,t}^{u^*} \right)^2 \frac{\mathbb{E}\left[ \|W_t\|_2 \right]}{\alpha_t \sigma_t} \right) L_{1,t+1}^{\widehat{V}} L_{0,t+1}^{\widehat{V}} (1-\lambda_t)^m \|y_1 - y_2\|_2 . \tag{67}
\end{aligned}
$$

With there estimates in hand, we are ready to prove the Lipschitz and gradient Lipschitz conditions of $V_t^{(m)}$. Note that

$$
\frac{(1-\alpha_t)}{\sqrt{\alpha_t}} \left( \nabla u_t^{(m)}(y) - \nabla s_t^{\mathrm{pre}}(y) \right) = \frac{\sigma_t^2}{\beta_t} \widehat{\mathcal{W}}_{t+1}^{(m)}(y) \mathcal{U}_t^{(m)}(y).
$$

Consequently, for any $y \in \mathbb{R}^d$, we deduce from (53), (54), (58) and (61) that

$$
\begin{aligned}
\left\| \nabla V_t^{(m)}(y) \right\|_2 &\leq \left\| \mathcal{S}_t(y)^\top \widehat{\mathcal{G}}_{t+1}^{(m)}(y) \right\|_2 + \frac{\sigma_t^2}{\beta_t} \left\| \widehat{\mathcal{W}}_{t+1}^{(m)}(y) \right\|_2 \left\| \mathcal{U}_t^{(m)}(y) \right\|_2 \left\| \widehat{\mathcal{G}}_{t+1}^{(m)}(y) - \widehat{\mathcal{G}}_{t+1}^{(m-1)}(y) \right\|_2 \\
&\leq \frac{1}{\sqrt{\alpha_t}} (1 + (1-\alpha_t) L_{0,t}^s) L_{0,t+1}^{\widehat{V}} + \frac{\sigma_t^2}{\beta_t} L_{1,t+1}^{\widehat{V}} \left( \frac{1}{\sqrt{\alpha_t}} \left( 1 + (1-\alpha_t) L_{0,t}^{u^*} \right) \right) (1-\lambda_t)^m L_{0,t+1}^{\widehat{V}} \\
&\leq \frac{1}{\sqrt{\alpha_t}} (1 + (1-\alpha_t) L_{0,t}^s) L_{0,t+1}^{\widehat{V}} + \frac{L_{0,t+1}^{\widehat{V}}}{\sqrt{\alpha_t}} \left( 1 + (1-\alpha_t) L_{0,t}^{u^*} \right) (1-\lambda_t)^{m+1}.
\end{aligned}
$$

Finally, for any $y_1$ and $y_2$ in $\mathbb{R}^d$, it follows from (59) and (67) that

$$
\begin{aligned}
&\left\| \nabla V_t^{(m)}(y_1) - \nabla V_t^{(m)}(y_2) \right\|_2 \\
&\leq \left\| \mathcal{S}_t(y_1)^\top \widehat{\mathcal{G}}_{t+1}^{(m)}(y_1) - \mathcal{S}_t(y_2)^\top \widehat{\mathcal{G}}_{t+1}^{(m)}(y_2) \right\|_2 \\
&\quad + \frac{\sigma_t^2}{\beta_t} \left\| \left( \widehat{\mathcal{W}}_{t+1}^{(m)}(y_1) \mathcal{U}_t^{(m)}(y_1) \right)^\top \left( \widehat{\mathcal{G}}_{t+1}^{(m)}(y_1) - \widehat{\mathcal{G}}_{t+1}^{(m-1)}(y_1) \right) \right. \\
&\qquad\qquad \left. - \left( \widehat{\mathcal{W}}_{t+1}^{(m)}(y_2) \mathcal{U}_t^{(m)}(y_2) \right)^\top \left( \widehat{\mathcal{G}}_{t+1}^{(m)}(y_2) - \widehat{\mathcal{G}}_{t+1}^{(m-1)}(y_2) \right) \right\|_2 \\
&\leq \left( \frac{1}{\alpha_t} (1 + (1-\alpha_t) L_{0,t}^s)(1 + (1-\alpha_t) L_{0,t}^{u^*}) L_{1,t+1}^{\widehat{V}} + \frac{1-\alpha_t}{\sqrt{\alpha_t}} L_{1,t}^s L_{0,t+1}^{\widehat{V}} \right) \|y_1 - y_2\|_2 \\
&\quad + \frac{\sigma_t^2}{\beta_t} L_{1,t+1}^{\widehat{V}} \left( \frac{1-\alpha_t}{\sqrt{\alpha_t}} L_{1,t}^{u^*} + (m+2) \left( 1 + (1-\alpha_t) L_{0,t}^{u^*} \right)^2 \frac{\mathbb{E}\left[ \|W_t\|_2 \right]}{\alpha_t \sigma_t} \right) L_{0,t+1}^{\widehat{V}} (1-\lambda_t)^m \|y_1 - y_2\|_2 \\
&\leq \left( \frac{1}{\alpha_t} (1 + (1-\alpha_t) L_{0,t}^s)(1 + (1-\alpha_t) L_{0,t}^{u^*}) L_{1,t+1}^{\widehat{V}} + \frac{1-\alpha_t}{\sqrt{\alpha_t}} L_{1,t}^s L_{0,t+1}^{\widehat{V}} \right. \\
&\quad \left. + \left( \frac{1-\alpha_t}{\sqrt{\alpha_t}} L_{1,t}^{u^*} + (m+2) \left( 1 + (1-\alpha_t) L_{0,t}^{u^*} \right)^2 \frac{\mathbb{E}\left[ \|W_t\|_2 \right]}{\alpha_t \sigma_t} \right) L_{0,t+1}^{\widehat{V}} (1-\lambda_t)^{m+1} \right) \|y_1 - y_2\|_2 ,
\end{aligned}
$$

which finishes the proof. □

Our next result characterizes the error of the control sequence obtained from the one-step update rule (44).

**Lemma C.2** (One-step error analysis on controls). *Assume the same assumptions as in Lemma C.1. Moreover, suppose there is a constant $E_{t+1}$ such that for all $y$ that*

$$\left\| \left( \nabla \widehat{V}_{t+1} - \nabla V_{t+1}^* \right)(y) \right\|_2 \leq E_{t+1}.$$

*Let $u_t^{(m)}$ and $V_t^{(m)}$ be defined as in (44) and (45), respectively. Then it holds that for all $y \in \mathbb{R}^d$ that*

$$\left\| u_t^{(m)}(y) - u_t^*(y) \right\|_2 \leq \left( (1 - \lambda_t)^n L_{0,t+1}^{V^*} + \lambda_t^{-1} E_{t+1} \right) \frac{\sqrt{\alpha_t}(1 - \lambda_t)}{(1 - \alpha_t) L_{1,t+1}^{V^*}}.$$

*Proof.* Define the Bellman optimality operator $\mathcal{T}_t$ at time $t$ as

$$(\mathcal{T}_t u)(y) := s_t^{\text{pre}}(y) + \frac{\sqrt{\alpha_t}\sigma_t^2}{(1 - \alpha_t)\beta_t} \mathbb{E} \left[ \nabla V_{t+1}^* \left( \frac{1}{\sqrt{\alpha_t}} (y + (1 - \alpha_t)u(y)) + \sigma_t W_t \right) \right].$$

Moreover, define the approximate Bellman operator $\widehat{\mathcal{T}}_t$ as

$$(\widehat{\mathcal{T}}_t u)(y) := s_t^{\text{pre}}(y) + \frac{\sqrt{\alpha_t}\sigma_t^2}{(1 - \alpha_t)\beta_t} \mathbb{E} \left[ \nabla \widehat{V}_{t+1} \left( \frac{1}{\sqrt{\alpha_t}} (y + (1 - \alpha_t)u(y)) + \sigma_t W_t \right) \right].$$

The update rule (44) implies $u_t^{(m+1)} = \widehat{\mathcal{T}}_t u_t^{(m)}$. Together with the optimality condition $u_t^* = \mathcal{T}_t u_t^*$, we obtain that

$$\begin{aligned}
\left\| u_t^{(m+1)}(y) - u_t^*(y) \right\|_2 &= \left\| (\widehat{\mathcal{T}}_t u_t^{(m)})(y) - (\mathcal{T}_t u_t^*)(y) \right\|_2 \\
&\leq \left\| (\widehat{\mathcal{T}}_t u_t^{(m)})(y) - (\mathcal{T}_t u_t^{(m)})(y) \right\|_2 + \left\| (\mathcal{T}_t u_t^{(m)})(y) - (\mathcal{T}_t u_t^*)(y) \right\|_2.
\end{aligned} \tag{68}$$

Note that $\mathcal{T}_t$ is a contraction operator. Indeed, for any $u, v : \mathbb{R}^d \to \mathbb{R}^d$ we have

$$\begin{aligned}
\|(\mathcal{T}_t u)(y) - (\mathcal{T}_t v)(y)\|_2 &\leq \frac{\sqrt{\alpha_t}\sigma_t^2}{(1 - \alpha_t)\beta_t} \mathbb{E} \Bigg[ \Bigg\| \nabla V_{t+1}^* \left( \frac{1}{\sqrt{\alpha_t}} (y + (1 - \alpha_t)u(y)) + \sigma_t W_t \right) \\
&\qquad\qquad - \nabla V_{t+1}^* \left( \frac{1}{\sqrt{\alpha_t}} (y + (1 - \alpha_t)v(y)) + \sigma_t W_t \right) \Bigg\|_2 \Bigg] \\
&\leq \frac{\sigma_t^2}{\beta_t} L_{1,t+1}^{V^*} \|u(y) - v(y)\|_2 \\
&\leq (1 - \lambda_t) \|u(y) - v(y)\|_2,
\end{aligned} \tag{69}$$

where we apply the Lipschitz condition of $\nabla V_{t+1}^*$ and the choice of $\beta_t$. Consequently, the second term in (68) is bounded with

$$\left\| (\mathcal{T}_t u_t^{(m)})(y) - (\mathcal{T}_t u_t^*)(y) \right\|_2 \leq (1 - \lambda_t) \left\| u_t^{(m)}(y) - u_t^*(y) \right\|_2. \tag{70}$$

Next, define

$$\mathcal{G}_{t+1}^{(m)}(y) := \mathbb{E} \left[ \nabla V_{t+1}^* \left( \frac{1}{\sqrt{\alpha_t}} \left( y + (1 - \alpha_t)u_t^{(m)}(y) \right) + \sigma_t W_t \right) \right],$$

and recall that

$$\widehat{\mathcal{G}}_{t+1}^{(m)}(y) = \mathbb{E} \left[ \nabla \widehat{V}_{t+1} \left( \frac{1}{\sqrt{\alpha_t}} \left( y + (1 - \alpha_t)u_t^{(m)}(y) \right) + \sigma_t W_t \right) \right].$$

For the first term in (68), note that

$$\left\| (\widehat{\mathcal{T}}_t u_t^{(m)})(y) - (\mathcal{T}_t u_t^{(m)})(y) \right\|_2 = \frac{\sqrt{\alpha_t}\sigma_t^2}{(1 - \alpha_t)\beta_t} \left\| \widehat{\mathcal{G}}_{t+1}^{(m)}(y) - \mathcal{G}_{t+1}^{(m)}(y) \right\|_2$$

$$\leq \frac{\sqrt{\alpha_t}\sigma_t^2}{(1-\alpha_t)\beta_t}\mathbb{E}\left[\left\|\left(\nabla\widehat{V}_{t+1} - \nabla V_{t+1}^*\right)(y')\right\|_2\right] \leq \frac{\sqrt{\alpha_t}\sigma_t^2}{(1-\alpha_t)\beta_t}E_{t+1}, \tag{71}$$

where $y' \sim \mathcal{N}\left(\frac{1}{\sqrt{\alpha_t}}\left(y + (1-\alpha_t)u_t^{(m)}\right)(y)\right), \sigma_t^2 I_d\right)$.

With (70) and (71), we can further bound (68) with

$$\left\|u_t^{(m+1)}(y) - u_t^*(y)\right\|_2 \leq \frac{\sqrt{\alpha_t}\sigma_t^2}{(1-\alpha_t)\beta_t}E_{t+1} + (1-\lambda_t)\left\|u_t^{(m)}(y) - u_t^*(y)\right\|_2. \tag{72}$$

Unrolling the recursion (72), we have

$$\left\|u_t^{(m)}(y) - u_t^*(y)\right\|_2 \leq (1-\lambda_t)^m\left\|u_t^{(0)}(y) - u_t^*(y)\right\|_2 + \lambda_t^{-1}\frac{\sqrt{\alpha_t}\sigma_t^2}{(1-\alpha_t)\beta_t}E_{t+1}$$

$$\leq \left((1-\lambda_t)^m L_{0,t+1}^{V^*} + \lambda_t^{-1}E_{t+1}\right)\frac{\sqrt{\alpha_t}\sigma_t^2}{(1-\alpha_t)\beta_t}$$

$$\leq \left((1-\lambda_t)^m L_{0,t+1}^{V^*} + \lambda_t^{-1}E_{t+1}\right)\frac{\sqrt{\alpha_t}(1-\lambda_t)}{(1-\alpha_t)L_{1,t+1}^{V^*}},$$

where we apply the condition that $1 - \frac{\sigma_t^2}{\beta_t}L_{1,t+1}^{V^*} \geq \lambda_t$. This finishes the proof. $\qquad\square$

The last lemma in this section is on the error analysis of $\nabla V_t^{(m)}$.

**Lemma C.3** (One-step error analysis on the gradient of value function). *Assume the same assumptions as in Lemma C.2. Then it holds that for all $y \in \mathbb{R}^d$,*

$$\left\|\nabla V_t^{(m)}(y) - \nabla V_t^*(y)\right\|_2 \leq C_{1,t}E_{t+1} + C_{2,t}(1-\lambda_t)^{m+1}, \tag{73}$$

*where*

$$C_{1,t} := \frac{1 + (1-\alpha_t)L_{0,t}^s}{\lambda_t\sqrt{\alpha_t}}, \quad \text{and} \quad C_{2,t} := \frac{\left(1 + (1-\alpha_t)L_{0,t}^{u^*}\right)}{\sqrt{\alpha_t}}L_{0,t+1}^{\widehat{V}} + \frac{\left(1 + (1-\alpha_t)L_{0,t}^s\right)}{\sqrt{\alpha_t}}L_{0,t+1}^{V^*}.$$

*Proof.* Recall the definition that $\widehat{\mathcal{G}}_{t+1}^{(m)}(y) = \mathbb{E}\left[\nabla\widehat{V}_{t+1}\left(\frac{1}{\sqrt{\alpha_t}}\left(y + (1-\alpha_t)u_t^{(m)}(y)\right) + \sigma_t W_t\right)\right]$. It follows from (43) and (57) that

$$\nabla V_t^{(m)}(y) - \nabla V_t^*(y) = \frac{1}{\sqrt{\alpha_t}}(I_d + (1-\alpha_t)\nabla s_t^{\mathrm{pre}}(y))^\top\left(\widehat{\mathcal{G}}_{t+1}^{(m)}(y) - \mathcal{G}_{t+1}(y)\right)$$

$$+ \frac{(1-\alpha_t)}{\sqrt{\alpha_t}}\left(\nabla u_t^{(m)}(y) - \nabla s_t^{\mathrm{pre}}(y)\right)^\top\left(\widehat{\mathcal{G}}_{t+1}^{(m)}(y) - \widehat{\mathcal{G}}_{t+1}^{(m-1)}(y)\right). \tag{74}$$

Observe that for any $m \geq 0$, it holds that

$$\left\|\widehat{\mathcal{G}}_{t+1}^{(m)}(y) - \mathcal{G}_{t+1}(y)\right\|_2 \leq \left\|\widehat{\mathcal{G}}_{t+1}^{(m)}(y) - \mathcal{G}_{t+1}^{(m)}(y)\right\|_2 + \left\|\mathcal{G}_{t+1}^{(m)}(y) - \mathcal{G}_{t+1}(y)\right\|_2, \tag{75}$$

where we recall $\mathcal{G}_{t+1}^{(m)}(y) = \mathbb{E}\left[\nabla V_{t+1}^*\left(\frac{1}{\sqrt{\alpha_t}}\left(y + (1-\alpha_t)u_t^{(m)}(y)\right) + \sigma_t W_t\right)\right]$. For the first term on the right hand side of (75), we notice the following inequality holds:

$$\left\|\widehat{\mathcal{G}}_{t+1}^{(m)}(y) - \mathcal{G}_{t+1}^{(m)}(y)\right\|_2 \leq \mathbb{E}\left[\left\|\left(\nabla\widehat{V}_{t+1} - \nabla V_{t+1}^*\right)(y')\right\|_2\right] \leq E_{t+1}, \tag{76}$$

where $y' \sim \mathcal{N}\left(\frac{1}{\sqrt{\alpha_t}}\left(y + (1-\alpha_t)u_t^{(m)}\right)(y)\right), \sigma_t^2 I_d\right)$. For the second term in (75), the Lipschitz continuity of $\nabla V_{t+1}^*$ implies

$$\left\|\mathcal{G}_{t+1}^{(m)}(y) - \mathcal{G}_{t+1}(y)\right\|_2 \leq L_{1,t+1}^{V^*}\frac{1-\alpha_t}{\sqrt{\alpha_t}}\left\|u_t^{(m)}(y) - u_t^*(y)\right\|_2. \tag{77}$$

Consequently, Eq. (75) becomes

$$\left\|\widehat{\mathcal{G}}_{t+1}^{(m)}(y) - \mathcal{G}_{t+1}(y)\right\|_2 \le E_{t+1} + L_{1,t+1}^{V^*} \frac{1-\alpha_t}{\sqrt{\alpha_t}} \left\|u_t^{(m)}(y) - u_t^*(y)\right\|_2. \tag{78}$$

With (53), (54), (61) and (78), we further bound (74) with

$$\left\|\nabla V_t^{(m)}(y) - \nabla V_t^*(y)\right\|_2 \le C_{1,t}' \left(E_{t+1} + L_{1,t+1}^{V^*} \frac{1-\alpha_t}{\sqrt{\alpha_t}} \left\|u_t^{(m)}(y) - u_t^*(y)\right\|_2\right) + C_{2,t}'(1-\lambda_t)^{m+1},$$

where the coefficients are given by

$$C_{1,t}' = \frac{1}{\sqrt{\alpha_t}}(1 + (1-\alpha_t)L_{0,t}^s), \text{ and } C_{2,t}' = \frac{1 + (1-\alpha_t)L_{0,t}^{u^*}}{\sqrt{\alpha_t}} L_{0,t+1}^{\widehat{V}}.$$

Finally, we apply Lemma C.2 to obtain

$$\begin{aligned}
\left\|\nabla V_t^{(m)}(y) - \nabla V_t^*(y)\right\|_2 &\le C_{1,t}' \left(E_{t+1} + L_{1,t+1}^{V^*} \frac{\sigma_t^2}{\beta_t}\left((1-\lambda_t)^m L_{0,t+1}^{V^*} + \lambda_t^{-1} E_{t+1}\right)\right) + C_{2,t}'(1-\lambda_t)^{m+1} \\
&\le C_{1,t}' \left(E_{t+1} + (1-\lambda_t)\left((1-\lambda_t)^m L_{0,t+1}^{V^*} + \lambda_t^{-1} E_{t+1}\right)\right) + C_{2,t}'(1-\lambda_t)^{m+1} \\
&\le C_{1,t} E_{t+1} + C_{2,t}(1-\lambda_t)^{m+1},
\end{aligned}$$

where

$$C_{1,t} = \lambda_t^{-1} C_{1,t}', \text{ and } C_{2,t} = C_{2,t}' + C_{1,t}' L_{0,t+1}^{V^*}.$$

Therefore, we finish the proof. □

Now we are ready to prove Theorem 3.1 with the results in Lemmas C.1, C.2 and C.3.

*Proof of Theorem 3.1.* It is straightforward to check that $L_{1,t}^{\widehat{V}} \ge L_{1,t}^{V^*}$ for all $t \le T$. Thus, the choice of $\beta_t$ guarantees that Theorem 2.8 holds. To prove the theorem, we begin with the error $\left\|\nabla V_t^{(m_t)}(y) - \nabla V_t^*(y)\right\|_2$. Let $t < T$ be fixed. To apply Lemma C.3, we choose $\widehat{V}_{t'} = V_{t'}^{(m_{t'})}$ for all $t' > t$. Unrolling recursion (73) and use the fact that $V_T^{(m_T)}(y) = r(y) = V_T^*(y)$, we obtain that

$$\begin{aligned}
\left\|\nabla V_t^{(m_t)}(y) - \nabla V_t^*(y)\right\|_2 &\le \prod_{k=t}^{T} C_{1,k} \left\|\nabla V_T^{(m_T)}(y) - V_T^*(y)\right\|_2 + \sum_{k=t}^{T-1}\left(\prod_{\ell=t}^{k-1} C_{1,\ell}\right) C_{2,k}(1-\lambda_k)^{m_k+1} \\
&= \sum_{k=t}^{T-1}\left(\prod_{\ell=t}^{k-1} C_{1,\ell}\right) C_{2,k}(1-\lambda_k)^{m_k+1}.
\end{aligned}$$

Next, we apply Lemma C.2 to have

$$\begin{aligned}
\left\|u_t^{(m_t)}(y) - u_t^*(y)\right\|_2 &\le \left((1-\lambda_t)^{m_t} L_{0,t+1}^{V^*} + \lambda_t^{-1}\left\|\nabla V_{t+1}^{m_{t+1}}(y) - \nabla V_{t+1}^*(y)\right\|_2\right) \frac{\sqrt{\alpha_t}(1-\lambda_t)}{(1-\alpha_t)L_{1,t+1}^{V^*}} \\
&\le \left((1-\lambda_t)^{m_t} L_{0,t+1}^{V^*} + \lambda_t^{-1} \sum_{k=t+1}^{T-1}\left(\prod_{\ell=t+1}^{k-1} C_{1,\ell}\right) C_{2,k}(1-\lambda_k)^{m_k+1}\right) \frac{\sqrt{\alpha_t}(1-\lambda_t)}{(1-\alpha_t)L_{1,t+1}^{V^*}},
\end{aligned}$$

which finishes the proof.

□

