# OpenReview forum: "Stochastic Control for Fine-tuning Diffusion Models: Optimality, Regularity, and Convergence"
_ICML.cc/2025/Conference — ICML 2025 poster_

### Official Review · Reviewer_1H7D · 2025-02-24

**Overall Recommendation:** 3

**Summary:**

This paper proposes a discrete-time stochastic control framework with linear dynamics and KL regularization for fine-tuning diffusion models. It establishes well-posedness, proves the regularity of the optimal value function, and develops a policy iteration algorithm (PI-FT) with guaranteed regularity and stability. The framework extends to parametric settings for efficient high-dimensional implementation.

**Claims And Evidence:**

The claims in this manuscript are supported by clear and convincing evidence, such as
1. The illustration of the connection between DDPM and stochastic control
2. The regularity and well-posedness of the optimal value function and the optimal control policy
3. The framework/algorithm for fine-tuning diffusion models (PI-FT) and the analysis for its convergence
4. linear parameterization to enable practical and efficient update for above algorithm

**Essential References Not Discussed:**

The paper provides a comprehensive discussion of related works. All essential references needed to understand its key contributions are adequately cited and analyzed.

**Experimental Designs Or Analyses:**

Unfortunately, the method proposed in this paper has not yet been demonstrated on a specific dataset or task. Incorporating experimental results would further validate the approach and enhance its credibility.

**Methods And Evaluation Criteria:**

Both the regularity & well-posedness and the convergence analysis give the proposed method a strong theoretical foundation. However none of the above theoretical derivation results and convergence of the algorithm have been realized on a particular dataset and a specific task. I think adding some experimental results will make this approach more convincing.

**Other Comments Or Suggestions:**

Including experimental results would strengthen the paper by demonstrating the practical effectiveness of the proposed approach. Empirical validation would also make the theoretical contributions more compelling and applicable to real-world scenarios.

**Other Strengths And Weaknesses:**

I have to say that the theory in this paper is very solid. However, the lack of empirical results is also the biggest drawback of this paper.

**Questions For Authors:**

1. Line90-Line96 in introduction section, authors mention that recent empirical results provide compelling evidence to support their framework. Could authors show more details about the comparison and similarity of their methods and your method.
2. Actually, authors can show the convergence and the regularity of the optimal value function & the optimal control policy of your approach based on above mentioned empirical settings.
3. The parameterization trick in proposed algorithm a linear convergence rate. The linear convergence rate means convergence faster and obtaining better performance with same time/resources? I am curious could the conditions in above lemmas and theorems be realized in empirical settings. External experiment results to prove this point is better.

**Relation To Broader Scientific Literature:**

Recent empirical results provide compelling evidence to support proposed framework, which implements reinforcement learning (RL) or control-based fine-tuning algorithms closely aligned with their approach. These empirical studies have demonstrated the practical efficacy of KL-regularized control formulations in fine-tuning diffusion models.

**Theoretical Claims:**

This paper provides strong theoretical foundations for fine-tuning diffusion models, with rigorous derivations supporting its key claims. It establishes a clear connection between Denoising Diffusion Probabilistic Models (DDPM) and stochastic control, ensuring a principled approach to fine-tuning. The authors prove the well-posedness and regularity of the optimal value function and control policy, guaranteeing stability. They develop a policy iteration algorithm (PI-FT) and provide a detailed convergence analysis. Additionally, the framework extends to a parametric setting with linear parameterization, enabling efficient and practical updates for high-dimensional applications.

---

> ### Author Rebuttal · Authors · 2025-03-31
>
> Thank you for your feedback. We are glad that you found our theoretical results strong and rigorous. Below please find our response to your questions.
> ```
> I have to say that the theory in this paper is very solid. However, the lack of empirical results is also the biggest drawback of this paper.
> ```
> Thank you for your suggestion. To demonstrate the practical efficiency and effectiveness of the proposed PI-FT method, we conduct thorough numerical experiments, which are included in https://anonymous.4open.science/r/ICML-rebuttal-741C/figures.pdf.
>
> **Experiment Set-up.** We fine-tune Stable Diffusion v1.5 for text-to-image generation using LoRA and ImageReward. Our prompts—“A green-colored rabbit,” “A cat and a dog,” “Four wolves in the park,” and “A dog on the moon”—assess color, composition, counting, and location. During fine-tuning, we generate 10 trajectories (50 transitions each) to calculate the gradient with 1K gradient steps (AdamW, LR = 3e-4), and KL regularization ($\beta= 0.01$).
>
> **Evaluation.** We first compare ImageReward scores of images generated by the pre-trained model, DPOK and PI-FT. To ensure a fair evaluation, we perform 10 gradient steps with LR = 1e-5 per sampling step in DPOK. Each gradient step is computed using 50 randomly sampled transitions from the replay buffer. Consequently, 1K sampling steps and 10K gradient steps in DPOK result in a computational cost similar to that of PI-FT. As shown in Figure 1 in the link, PI-FT consistently **outperforms** both the pre-trained model and DPOK across all four prompts. Figure 2 further shows that PI-FT more accurately generates the correct number of wolves, and correctly places the dog on the moon. Additionally, PI-FT avoids mistakes such as depicting a rabbit with a green background as appeared in the pre-trained model and DPOK. The texture of the generated animals is also more natural-looking under PI-FT compared to other baselines.
>
> **Effect of KL regularization.** KL regularization is known to improve fine-tuning. We analyze its effect in the PI-FT method using the prompt “Four wolves in the park,” varying $\beta \in$ {0.01, 0.1, 1.0}. Figure 3 shows that the gradient norm decreases to 0 in all three settings, illustrating the convergence of our algorithm. In Figure 4, we see that the ImageReward score increases and eventually stabilizes when $\beta$ is small. In contrast, the score exhibits limited improvement with large $\beta$. This observation is consistent with Figure 5, where the KL divergence remains large throughout training for $\beta = 0.01$ and stays significantly at a smaller level for $\beta \in$ {0.1, 1.0}. Figure 6 illustrates that smaller $\beta$ produces images with nearly four wolves, while larger $\beta$ yields fewer. These results underscore the importance of the KL coefficient in our framework.
> ```
> … Could authors show more details about the comparison and similarity of their methods and your method.
> ```
> In the literature, DPOK and DDPO use generic RL algorithms such as PPO or REINFORCE to optimize the policy/score. Despite showing empirical promise on certain tasks, existing methods **overlook** the underlying structure of diffusion models, leaving significant room for improvement in efficiency and lacking any theoretical justification. In contrast, by fully utilizing the problem structure, we formulate fine-tuning as a KL-regularized stochastic control problem (with linear dynamics), and propose the PI-FT algorithm. This structure allows us to derive regularity and convergence guarantees, closing the gap left by prior works. We will revise lines 90-96 to clarify these points.
> ```
> ... show the convergence and the regularity of the optimal value function & the optimal control policy of your approach based on above mentioned empirical settings.
> ```
> In the newly added experiments, we do observe convergence of both the control policy and the value function during training, supporting our theoretical claims.
> ```
> … I am curious could the conditions in above lemmas and theorems be realized in empirical settings…
> ```
> Empirically, we observe numerical results consistent with our theoretical claims. In Figure 3 (in the link), we observe that the algorithm converges **linearly** and the gradient norm of the U-Net stabilizes after approximately 200 sampling steps. Additionally, we compare PI-FT and DPOK under the same computational budgets; as shown in Figure 1, PI-FT achieves consistently higher ImageReward scores across all four prompts. This performance suggests that our approach converges more efficiently in practice. Although our theory assumes Lipschitz conditions and a specific range of the regularization coefficient, our experiments demonstrate that convergence remains robust even for small $\beta = 0.01$ and relatively large learning rates.
>
> Thank you for your detailed feedback. We hope our response addresses your concerns. If so, we would appreciate it if you could reflect it in your evaluation of our paper.

---

> > ### Comment · Reviewer_1H7D · 2025-04-02
> >
> > The author's response helped clarify some of the confusion I had about this article. While the experimental results do not fully establish the validity of the proposed method, they serve as strong supporting evidence for the paper’s theoretical framework. I look forward to seeing more experimental results and comparisons of relevant evaluation metrics in the final version. I correspondingly improved my score.

---

### Official Review · Reviewer_7pRd · 2025-03-10

**Overall Recommendation:** 4

**Summary:**

This paper proposes a stochastic control framework for fine-tuning diffusion models. The key contribution is establishing theoretical properties such as well-posedness, regularity, and linear convergence of a proposed policy iteration algorithm.

**Claims And Evidence:**

The paper makes strong theoretical claims about the optimality, regularity, and linear convergence of their proposed algorithm. The theoretical analysis is mathematically rigorous with assumptions clearly presented.

**Essential References Not Discussed:**

Reference discussion is proper and abundant to me.

**Ethics Expertise Needed:**

["Legal Compliance (e.g., GDPR, copyright, terms of use)"]

**Experimental Designs Or Analyses:**

A significant weakness is the absence of any empirical validation. The paper does not include simulations or practical experiments demonstrating the behavior of Algorithm 1 or verifying the theoretical claims. In diffusion model fine-tuning, it is a common practice to test it on text-to-image models such as StableDiffusion v1.5. Pipelines for such implementations are also accessible and easy to work with. Can the authors provide clarifications why it is not conducted?

**Methods And Evaluation Criteria:**

Proper evaluation is missing: Algorithm 1 is defined and justified clearly but is purely theoretical; however, the practicality of the method is questionable. The algorithm’s iterative nature and associated computational overhead might limit its applicability in realistic high-dimensional problems, but no practical evaluation or even basic numerical simulations have been conducted.

**Other Comments Or Suggestions:**

I find following the flow of this work pretty easy and smooth. One suggestion would be: all theoretical results before Thm 2.8 are standard  in the literature, that is to say, all contents up until Page 5. Results such as Lemma 2.5 and Eq. 8 (many remarks are also well-known) have long been discovered, see [1], [2]. These standard results take up too much space in the main text. I would suggest improving the presentation, for example, simply referring to these works instead of repeating all the details in the main text. Otherwise, it is also possible to only present the most important parts in the main text but defer burdensome calculation details and remarks to the appendix in the next version.

More importantly, since this work is purely theoretical and does not have any kind of simulation studies. It is advised to spend more efforts in demonstrating the technical hardness of Thm 2.8 and Thm 3.1. It is confusing to me what the major difficulty is in presenting these results. It feels like Thm 2.8 is a direct calculation based on the Lipschitzness assumptions, which might not present enough technical novelty. I might be wrong, but can the authors comment on this?


[1] https://arxiv.org/pdf/2402.15194
[2] https://arxiv.org/abs/2403.06279

**Other Strengths And Weaknesses:**

As mentioned, Algorithm 1 appears overly idealized, and its practical utility in realistic scenarios remains highly questionable. In practical high-dimensional environment, taking expectations to obtain accurate value function can have difficulty. But the idealized algorithm overlooks such hardness.

**Questions For Authors:**

See above

**Relation To Broader Scientific Literature:**

The paper accurately positions itself within existing literature concerning fine-tuning diffusion models using reinforcement learning and stochastic control

**Theoretical Claims:**

I have roughly examined the theoretical analysis, particularly the convergence properties presented in Theorem 3.1 and Theorem 2.8. The proofs appear correct. (On the other hand, all results before Thm 2.8 are standard and well-known in the literature. I have concerns of ssuch presentation, see below)

---

> ### Author Rebuttal · Authors · 2025-03-31
>
> Thank you for your constructive feedback. We are delighted that you found our theoretical results rigorous and our presentation clear. Below is our point-to-point response to your comments:
> ```
> Proper evaluation is missing: Algorithm 1 is defined and justified clearly but is purely theoretical; however, the practicality of the method is questionable…
> ```
> ```
> A significant weakness is the absence of any empirical validation…
> ```
> ```
> As mentioned, Algorithm 1 appears overly idealized, and its practical utility in realistic scenarios remains highly questionable. In practical high-dimensional environment, taking expectations to obtain accurate value function can have difficulty…
> ```
> We included a new set of thorough experiments in this rebuttal. Please see https://anonymous.4open.science/r/ICML-rebuttal-741C/figures.pdf for the results and see response to Reviewer 1H7D for detailed explanations on the experiments.
> ```
> I find following the flow of this work pretty easy and smooth. One suggestion would be: all theoretical results before Thm 2.8 are standard in the literature, that is to say, all contents up until Page 5. Results such as Lemma 2.5 and Eq. 8 (many remarks are also well-known) have long been discovered, see [1], [2]. These standard results take up too much space in the main text. I would suggest improving the presentation, for example, simply referring to these works instead of repeating all the details in the main text. Otherwise, it is also possible to only present the most important parts in the main text but defer burdensome calculation details and remarks to the appendix in the next version.
> ```
> Thank you for your constructive feedback. To the best of our knowledge, we are the first to use KL divergence over transition dynamics (on path space) to control the deviation of the fine-tuned model from the pre-trained model, whereas formulations in the literature consider KL between the terminal state distributions. For this reason, the results before Theorem 2.8 are novel and valuable to readers although not technically significant. Therefore, we decide to follow your suggestion and defer calculation details and remarks to the appendix in the revised manuscript.
> ```
> …since this work is purely theoretical and does not have any kind of simulation studies. It is advised to spend more efforts in demonstrating the technical hardness of Thm 2.8 and Thm 3.1. It is confusing to me what the major difficulty is in presenting these results. It feels like Thm 2.8 is a direct calculation based on the Lipschitzness assumptions, which might not present enough technical novelty. I might be wrong, but can the authors comment on this?
> ```
> We appreciate the opportunity to clarify the technical novelty of Theorem 2.8 and Theorem 3.1.
>
> Regarding Theorem 2.8, the core challenge stems from the fact that Eq. (17) is an implicit equation, in which both sides involve the optimal control $u_t^*$. This prevents a direct application of Lipschitz assumptions on model parameters. In contrast,  we prove the Lipschitz property of $u_t^*$ through the detailed and nontrivial derivations in Line 726 - 736, where the choice of $\beta_t$ ensures the invertibility of the coefficient. Additionally, the differentiability of $u_t^*$ relies on the Gaussian smoothing effect; see Eq. (30) – (32), which is novel analysis to the RL literature. The Lipschitz condition of $\nabla u_t^*$ is especially non-trivial as it involves $\nabla^2 V_{t+1}^*$, which may not be Lipschitz. We address this using the integration by parts formula in Eq. (34), which is a key technical step. We will discuss these in the manuscript.
>
> In terms of Theorem 3.1, the convergence analysis of the PI-FT algorithm relies on **maintaining the regularity** of the value function throughout training, which is highly nontrivial. For example, prior work such as [3] proves convergence under the assumption that the value function remains regular during training (which is non-tractable); see Assumption 2 in [3]. In contrast, our result establishes the desirable regularity through a connection between $V_t^{(m)}$ and the optimal value function $V_t^*$.
>
> Thanks to the reviewer’s question, we will revise the manuscript to better emphasize the technical challenges and contributions underlying these results.
>
> \
> Thank you for your time and for your valuable feedback. Your suggestions definitely helped us improve the quality of this manuscript and highlight our contributions. We hope our response addresses your questions and clarifies the challenges in our work. We also hope that the effort we invested in conducting thorough numerical experiments demonstrates the promising practical performance of our proposed algorithm. If so, we would be grateful if you could reflect it in your evaluation of our paper.
>
> [3] Zhou, Mo, and Jianfeng Lu. "A policy gradient framework for stochastic optimal control problems with global convergence guarantee." arXiv preprint arXiv:2302.05816 (2023).

---

### Official Review · Reviewer_Yp8J · 2025-03-13

**Overall Recommendation:** 3

**Summary:**

The authors propose a discrete-time stochastic optimal control framework with linear dynamics and KL regularization to model the problem of fine-tuning of diffusion models. They analyze well-posedness and regularity of the control formulation, propose a novel algorithmic scheme based on policy iteration, and they analyze it showing linear convergence. Ultimately they introduce a parametric extension and analyze the convergence of an associated policy gradient method.

## update after rebuttal
The rebuttal and further discussion with the authors tackled properly some of my concerns and therefore I have increased the score from 2 to 3. Nonetheless, I still believe that some of the weaknesses mentioned in my review and further comments do still hold. In particular, I am not convinced by the authors justification for the current experimental evaluation proposed within the rebuttal. Although it certainly makes the work more complete than before, it arguably does not present a clear comparison with common methods in this space and might lead to a distorted/limited viewpoint/evaluation of the presented methods.

**Claims And Evidence:**

Yes

**Essential References Not Discussed:**

The work seems to properly mention all relevant references.

**Experimental Designs Or Analyses:**

There are no experiments.

**Methods And Evaluation Criteria:**

The paper proposes arguably novel Policy Iteration and Policy Gradient schemes for fine-tuning of diffusion models but does not perform any experiment (e.g., comparing with existing methods). This is not inherently problematic, as theoretically relevant papers without any experiment are fine, but as later pointed out, I have some doubts regarding the relevance of the presented theoretical results. And not presenting experimental validations of the presented algorithm fundamentally renders impossible to evaluate and/or them besides the theoretical results.

**Other Comments Or Suggestions:**

1. Line 434, 'replies' should be 'relies', I guess.
2. Line 397, 'the' is repeated twice.

**Other Strengths And Weaknesses:**

STRENGTHS:
1. The paper focuses on a very timely and relevant problem.
2. The authors analyze well-posedness and regularity properties of a practically relevant control problem, which I regard as a positive contribution.
3. The authors present an arguably new algorithms based on policy iteration and policy gradient for fine-tuning of diffusion models and an extension to parametric controls.
4. Formal derivations showing linear convergence guarantees of such algorithms are presented.

WEAKENESSES:
Unfortunately, I found the paper quite unclear in being able to distinguish between the policy iteration and policy gradient schemes presented, the limitations of each, and whether some limitations are purely of the theoretical analysis, or are actual algorithmic limitations. I list several concerns and doubts in the following.
1. It is not clear to me if the policy iteration algorithm presented is supposed to work only on discrete-spaces, or, more precisely, how the control $u$ is represented within that algorithm. What settings does this algorithm actually cover? I am asking this since Sec. 4 seems to extend the control representation to be parametrized, but in doing so it seems to propose another algorithm. Did I misinterpret something?
2. Sec. 4 seems to propose an algorithm for continuous spaces where the control is represented via a linear parametrization. While a linear approximation would make sense to represent the forward process drift, this seems an excessive approximation for the reverse process control, which should have an order of complexity comparable to a score network. It seems to me that this object would be non-linear even in the easiest examples possible. Hence the question: are there some relevant examples (even simple) where this assumption holds?
3. At times it is very unclear whether the algorithms are introduced within this work (e.g., the policy gradient one), or if this work aims to only perform theoretical analysis on them. Please clarify.


Not presenting experimental evaluations renders impossible to evaluate the practical relevance of proposed algorithmic ideas. Hence I have to evaluate only the theoretical contributions.

**Questions For Authors:**

Please answer clearly regarding all 3 points mentioned within the weaknesses section. I am very open to changing my vote/decision based on these answers.

**Relation To Broader Scientific Literature:**

- To the best of my knowledge most works within this literature (of diffusion models fine-tuning) are fundamentally experimental. Overall I believe the authors of this work reported a very clear and honest representation of the current landscape of existing works. I find particularly interesting the focus on discrete-time and linear dynamics, which seems well motivated and renders it possible to connect the problem to common formulations in control theory. Having said so, I have doubts regarding the relevance of the theoretical results derived expressed within the weaknesses section.

**Theoretical Claims:**

Only limited parts of Lemmas B.2 and B.3.

---

> ### Author Rebuttal · Authors · 2025-03-31
>
> Thank you for your detailed feedback. We are glad to receive your positive feedback on our theoretical contributions and that you found our placement of the work clear and honest. Below please find our response to your questions.
>
> ```
> It is not clear to me if the policy iteration algorithm presented is supposed to work only on discrete-spaces, or, more precisely, how the control $u$ is represented within that algorithm. What settings does this algorithm actually cover? I am asking this since Sec. 4 seems to extend the control representation to be parametrized, but in doing so it seems to propose another algorithm. Did I misinterpret something?
> ```
> While we consider a **discrete-time** model, the PI-FT algorithm is formulated on continuous state and action spaces, with both state and control taking values in $\mathbb{R}^d$. This setting captures the essence of fine-tuning and indeed disparts us from the usual RL literature. Specifically, we focus on Markovian control policy $u_t: \mathbb{R}^d \to \mathbb{R}^d$, and our algorithm iteratively updates $u_t(y)$ for each $y \in \mathbb{R}^d$. Section 4 introduced a linear parametrization of the control policy to enable an efficient implementation in practice. This parameterized setting should be viewed as a special version of the PI-FT algorithm, rather than a separate algorithm. Moreover, we believe the theoretical results regarding regularity and convergence established in Section 3 apply to this parameterized setting and a roadmap has been provided in Section 4. Please let us know if this explanation clarifies your concerns.
>
> ```
> Sec. 4 seems to propose an algorithm for continuous spaces where the control is represented via a linear parametrization. While a linear approximation would make sense to represent the forward process drift, this seems an excessive approximation for the reverse process control, which should have an order of complexity comparable to a score network. It seems to me that this object would be non-linear even in the easiest examples possible. Hence the question: are there some relevant examples (even simple) where this assumption holds?
> ```
> We refer to the parameterization in Section 4 as linear in the sense that the control policy is a linear combination, through parameter $K_t$, of (nonlinear) basis functions. Hence the resulting control can be possibly nonlinear in the state variable $y$. This is a flexible, powerful, and tractable parameterization often used in control and RL, where depending on the basis, it can have high expressivity. In particular, our parameterization includes a wide class of score approximators such as random features, kernel methods and even overparameterized neural networks under the NTK regime. Hence, despite being linear in parameter $K_t$, the policy class can be sufficiently rich to approximate highly nonlinear score functions. Moreover, our experimental results (which we discuss below) suggest our principled framework remains valid for generic neural networks. We will clarify these points in our revision.
>
> ```
> At times it is very unclear whether the algorithms are introduced within this work (e.g., the policy gradient one), or if this work aims to only perform theoretical analysis on them. Please clarify.
> ```
> Apologies for the confusion and thank you for raising this point. To clarify, both the policy iteration method (PI-FT) and its policy gradient variant are new and proposed in this work.  Existing methods such as DPOK and DDPO rely on generic RL developments such as PPO or REINFORCE and fail to fully leverage the fine-tuning structure. On the contrary, the specific setting considered, with linear dynamics and entropy-regularization, is tailored towards the development of efficient fine-tuning diffusion models. For this reason, the PI-FT algorithm and its parametric extension directly computes the policy gradient of a KL-regularized control objective. This principled design leads to a more efficient implementation in practice compared to prior works; see our experiments below. We will clarify all these points in our final revision. In particular, we will specifically highlight that both algorithms are proposed in this work and are our contribution.
>
> ```
> Not presenting experimental evaluations renders impossible to evaluate the practical relevance of proposed algorithmic ideas.
> ```
> We included a new set of thorough experiments in this rebuttal. Please see link https://anonymous.4open.science/r/ICML-rebuttal-741C/figures.pdf for the results and see response to Reviewer 1H7D for detailed explanations on the experiments.
>
> ```
> Line 434, 'replies' should be 'relies', I guess.
> Line 397, 'the' is repeated twice.
> ```
> Thank you for pointing out these typos. We will fix them in the revised manuscript.
>
>
> Finally, we would like to thank you for your constructive feedback. We hope our response answers your questions. If so, we would greatly appreciate it if you could reflect it in your evaluation of our paper.

---

> > ### Comment · Reviewer_Yp8J · 2025-04-02
> >
> > While the rebuttal clarifies some aspects regarding the theory, others are still unclear to me, in particular:
> >
> > 1) How should I interpret the sentence "This parameterized setting should be viewed as a special version of the PI-FT algorithm, rather than a separate algorithm"? It seems to me that this setting is formally tackled with another algorithm in Sec. 4 (namely the PG scheme). What am I missing?
> >
> > 2) While I understand in general the idea of learning a complex state representation and then considering a linear dynamics for the sake of analysis, in the case of diffusion modeling this seems to me particularly controversial as to my understanding they rely on the idea of defining a very complex vector field on the original space in order to implicitly represent complex distributions on the same space. This seems the case also when the diffusion process acts on a learned latent space. While I would understand considering a linear or kernelized approximation of the score/dynamics for algorithmic sake, I am not sure how much this is a fair abstraction for theoretical understanding in this context. Why would it be?
> >
> > Moreover, my main concern regarding this work is the following: I would understand building a theoretical analysis for practically successful methods, but the algorithms introduced here are novel and therefore would require experimental comparison with relevant methods. In the context of entropy-regularized control for fine-tuning, there are already several works with successful algorithms based on continuous-time control, e.g., [1,2] among others. Hence the question: why DPOK, which does not seem to rely on the classic duality for KL-regularized control/RL would be a meaningful baseline compared to these algorithms that rely on entropy/KL-regularized control?
> >
> > [1] Uehara et al., Feedback efficient online fine-tuning of diffusion models. ICML 2024.
> > [2] Domingo-Enrich et al., Adjoint matching: Fine-tuning flow and diffusion generative models with memoryless stochastic optimal control. ICLR 2025.

---

> > > ### Author Response · Authors · 2025-04-03
> > >
> > > We thank the reviewer for constructive feedback. Below is our response to your concerns.
> > > ```
> > >  "This parameterized setting should be viewed as a special version of the PI-FT, rather than a separate algorithm"? … this setting is formally tackled with another algorithm in Sec. 4 (namely the PG scheme).
> > > ```
> > > Apologies for the confusion and thank you for the opportunity to clarify. The **parameterized setting** in our earlier response refers to the PG scheme in Section 4. We would like to emphasize that Section 4 does not introduce a new algorithm rather presents a practically implementable **realization/actualization** of the PI-FT algorithm developed in Section 3. This realization is closely connected to the PI-FT in two fronts: First, the PG scheme implements the key ideas of PI-FT in a computationally tractable manner by restricting the control policy to a parameterized function class, especially for high-dimensional problems. Second, the convergence of PI-FT serves as the essential building block of the convergence of the (parameterized) PG scheme. In particular, the regularity of the optimal value function and its preservation during training are crucial for the convergence analysis. For this reason, we first derive the suite of results for the PI-FT algorithm in Section 3 and then introduce the PG scheme in Section 4. Building on the results derived in Section 3, we are nearly equipped with the theoretical results for the PG algorithm, with Section 4 providing a clear roadmap for the remaining (straightforward) steps.
> > >
> > > We will clarify this point in our revision. Please let us know if this explanation clarifies your concerns.
> > >
> > > ```
> > > …a linear dynamics for the sake of analysis,...diffusion process acts on a learned latent space…a linear or kernelized approximation…a fair abstraction…
> > > ```
> > > Thank you for the opportunity to clarify two types of linearity in our work.
> > >
> > > First, the linearity of the dynamics is not introduced for analytical convenience, but as a natural consequence of the DDPM backward process, which has been widely used in practice. In the pre-trained model, the DDPM sampling dynamics presented in Eq. (1) of our manuscript are generally non-linear in the state $Y_t^{\rm pre}$ due to the presence of the learned score function $s_t^{\rm pre}$. However, once the pre-trained score is replaced by a control variable, the resulting sampling dynamics become **linear in both state and control** with additive Gaussian noise (while **control can still be a nonlinear function in states**). This hidden **linear structure** is the key insight from practical fine-tuning applications that enables our effective control formulation. Although the idea of treating score as an action has appeared in prior work, we are the first to leverage this linear structure to design an efficient algorithm and establish convergence guarantees in the context of fine-tuning diffusion models. Empirical results further validate the efficiency and practicality of our framework.
> > >
> > > Moreover, the use of linear parameterization for the control policy is motivated from two aspects: First, the linear policy (in parameters) has been widely used in the RL and control literature. As mentioned in our earlier response, while the policy is linear in parameters, it remains nonlinear in the state due to expressive feature mappings, enabling it to capture a broad class of complex score functions. Second, the control policy can also be parameterized using (an overparameterized) neural network, which behaves similar to a linear model in the NTK regime [3]. We are confident that the analysis can be carried through under such neural network parameterization as well.
> > >
> > > We will clarify these points in our revision.
> > > ```
> > > … experimental comparison…e.g., [1,2] among others. Hence the question: why DPOK, which does not seem to rely on the classic duality for KL-regularized control/RL…
> > > ```
> > >
> > > We would like to clarify that DPOK [4] **does** have KL regularization; see sections 4.1 and 5.3 of [4]. Moreover, our work is based on **discrete-time** DDPM dynamics, which are natural settings in practical implementations of diffusion models. While prior works such as [1,2] study KL-regularized control problems in continuous time, their formulations and the additional discretizations may not align with the actual diffusion sampling processes, making it difficult to have a fair comparison to our setting. In contrast, DPOK studies the same discrete-time framework, which is a more suitable empirical baseline for our experiments. We will also include additional experiments in the revision.
> > >
> > > We hope our response addresses your additional concerns. If so, we would greatly appreciate it if you could reflect it in your evaluation of our paper.
> > >
> > >
> > > [3] Han, Y. et al.  Neural Network-Based Score Estimation in Diffusion Models: Optimization and Generalization. ICLR 2024.
> > >
> > > [4] Fan, Y. et al. DPOK: Reinforcement learning for fine-tuning text-to-image diffusion models. NeurIPS 2023.

---

### Decision · Program_Chairs · 2025-05-01

**Decision:**

Accept (poster)

**Comment:**

All reviewers appreciate the theoretical contribution of this paper. I also think in that way as well. However, I suggest citing related works in the proper place. For example, equations [7] and [8] in diffusion models were very explicitly derived in the following with a similar title paper:

Uehara, M., Zhao, Y., Black, K., Hajiramezanali, E., Scalia, G., Diamant, N. L., ... & Levine, S. (2024). Fine-tuning of continuous-time diffusion models as entropy-regularized control. arXiv preprint arXiv:2402.15194.

or Domingo-Enrich et al. (2024).